# Local gate control of Mott metal-insulator transition in a 2D metal-organic framework

Benjamin Lowe [1,2,6], Bernard Field [1,2,6], Jack Hellerstedt [1,2], Julian Ceddia [1,2], Henry L. Nourse [3], Ben J. Powell [4] ✉, Nikhil V. Medhekar [2,5] ✉ & Agustin Schiffrin [1,2] ✉

Electron-electron interactions in materials lead to exotic many-body quantum phenomena, including Mott metal-insulator transitions (MITs), magnetism, quantum spin liquids, and superconductivity. These phases depend on electronic band occupation and can be controlled via the chemical potential. Flat bands in two-dimensional (2D) and layered materials with a kagome lattice enhance electronic correlations. Although theoretically predicted, correlated-electron Mott insulating phases in monolayer 2D metal-organic frameworks (MOFs) with a kagome structure have not yet been realised experimentally. Here, we synthesise a 2D kagome MOF on a 2D insulator. Scanning tunnelling microscopy (STM) and spectroscopy reveal a MOF electronic energy gap of ~200 meV, consistent with dynamical mean-field theory predictions of a Mott insulator. Combining template-induced (via work function variations of the substrate) and STM probe-induced gating, we locally tune the electron population of the MOF kagome bands and induce Mott MITs. These findings enable technologies based on electrostatic control of many-body quantum phases in 2D MOFs.

Strong electronic correlations arise in a material at specific electron fillings of its bands, provided that the on-site Coulomb repulsion (characterised by the Hubbard energy, $U$) is of the order of, or larger than, the bandwidth, $W$. These electronic correlations can result in a wide range of exotic many-body quantum phases. Examples include correlated insulating phases, quantum spin liquids, correlated magnetism, and superconductivity—phenomena that have been realised in monolayer transition metal-dichalcogenides[1–6], twisted few-layer graphene[7,8], inorganic kagome crystals[9–11], and organic charge transfer salts[12–14].

Tuning of the chemical potential via electrostatic gating can allow for control over such band electron filling, enabling reversible switching between correlated phases[7]. This makes these systems amenable to integration as active materials in voltage-controlled devices, offering enticing prospects for applications in electronics, spintronics, and information processing and storage[12,15].

Two-dimensional (2D) materials have emerged as particularly promising candidates for realising strongly correlated phenomena as the absence of interlayer hopping and screening can contribute to decreasing $W$ and increasing $U$[4]. Additionally, some 2D crystal geometries—such as the kagome structure—give rise to intrinsic flat electronic bands[16,17]. When these extremely narrow bands are half-filled, even weak Coulomb repulsion can open an energy gap and give rise to a Mott insulating phase[12]. Away from half-filling, the gap closes, and the system becomes metallic.

Metal-organic frameworks (MOFs) are a broad class of materials whose properties are highly tunable through careful selection of constituent organic molecules and metal atoms[18]. There has been

[1]School of Physics and Astronomy, Monash University, Clayton, VIC, Australia. [2]ARC Centre of Excellence in Future Low-Energy Electronics Technologies, Monash University, Clayton, VIC, Australia. [3]Quantum Information Science and Technology Unit, Okinawa Institute of Science and Technology Graduate University, Onna-son, Okinawa, Japan. [4]School of Mathematics and Physics, The University of Queensland, Brisbane, QLD, Australia. [5]Department of Materials Science and Engineering, Monash University, Clayton, VIC, Australia. [6]These authors contributed equally: Benjamin Lowe, Bernard Field. ✉e-mail: powell@physics.uq.edu.au; nikhil.medhekar@monash.edu; agustin.schiffrin@monash.edu

growing interest in 2D MOFs for their electronic properties[19–21]. In particular, layered 2D MOF structures have been recently shown to host strongly correlated superconductivity[22]. Monolayer 2D MOFs have attracted attention for their magnetism[23–25], including ferromagnetism resulting from exchange interaction between unpaired metal centre electrons[26]. However, despite theoretical predictions[27,28], correlated Mott phases have not yet been realised experimentally in monolayer 2D MOFs.

Here, we demonstrate local electrostatic control over a Mott metal-insulator-transition (MIT) in a single-layer 2D kagome MOF, in excellent agreement with theoretical predictions.

## Results

### A monolayer 2D MOF on an atomically thin insulator

We synthesised the monolayer MOF−consisting of 9,10-dicyanoanthracene (DCA) molecules coordinated to copper (Cu) atoms−on monolayer hexagonal boron nitride (hBN) on Cu(111) (see Methods for sample preparation). A scanning tunnelling microscopy (STM) image of a crystalline single-layer MOF domain grown seamlessly across the hBN/Cu(111) substrate is shown in Fig. 1a. We observe some defects within the MOF domain, as well as some DCA-only regions (discussed in Supplementary Note 11). The long-range modulation of the MOF STM apparent height follows the hBN/Cu(111) moiré pattern, which arises due to a mismatch between the hBN and Cu(111) lattices (giving rise to pore, P, and wire, W, regions−see upper inset)[29–31]. This moiré pattern has been shown to affect the electronic properties of adsorbates[31], including one previous example of a MOF[32].

The MOF is characterised by a hexagonal lattice (lattice constant: $2.01 \pm 0.06$ nm), with a unit cell including two Cu atoms (honeycomb arrangement; bright protrusions in Fig. 1b) and three DCA molecules (kagome arrangement, with protrusions at both ends of anthracene backbone in STM image in Fig. 1b), similar to previous reports (including on a decoupling graphene monolayer)[25,33–35].

We calculated the band structure of this monolayer DCA$_3$Cu$_2$ MOF on hBN/Cu(111) by density functional theory (DFT; with $U = 0$); Fig. 1d.

Projection of the Kohn−Sham wavefunctions onto MOF states shows the prototypical kagome energy dispersion with two Dirac bands and a flat band, consistent with prior theoretical calculations for the freestanding MOF[19,27,36]. This near-Fermi band structure has predominantly molecular DCA character, and is well described by a nearest-neighbour tight-binding (TB) model (see the corresponding density of electronic states, DOS, as a function of energy $E$ in Fig. 1e)[36]. The hBN monolayer, a 2D insulator with a bandgap >5 eV[31], prevents electronic hybridisation between the underlying Cu(111) surface and the 2D MOF[31]. This allows the MOF to preserve its intrinsic electronic properties, in contrast to previous findings on metal surfaces[25,35–37]. These $U = 0$ calculations predict that the MOF on hBN/Cu(111) is metallic, with some electron transfer from substrate-to-MOF leading to the chemical potential lying above the Dirac point, close to half-filling of the three kagome bands (Fig. 1d).

Strong electronic interactions have been theoretically predicted in DCA$_3$Cu$_2$[27], and a signature was recently detected experimentally[25]. We, therefore, calculated the many-body spectral function $A(E)$−analogous to the DOS in the non-interacting regime−of the freestanding MOF via dynamical mean-field theory (DMFT). In contrast to the TB model or DFT, DMFT explicitly captures local electronic correlations caused by the Hubbard energy $U$ (see Methods)[38–41]. In Fig. 1e, $A(E)$ is shown for $U = 0.65$ eV (consistent with previous experimental estimates[33]) and for a chemical potential that matches the DFT-predicted occupation of the kagome bands for the MOF on hBN/Cu(111) (see Supplementary Note 2 and Supplementary Note 23 for further DMFT calculations, including temperature dependence). We observe two broad peaks (lower and upper Hubbard bands) separated by an energy gap of ~200 meV, dramatically different from the non-interacting kagome DOS, and indicative of a Mott insulating phase[27,42].

### Observation of ~200 meV Mott energy gap

To experimentally probe the electronic properties of DCA$_3$Cu$_2$/hBN/Cu(111), we conducted differential conductance (d$I$/d$V$) scanning

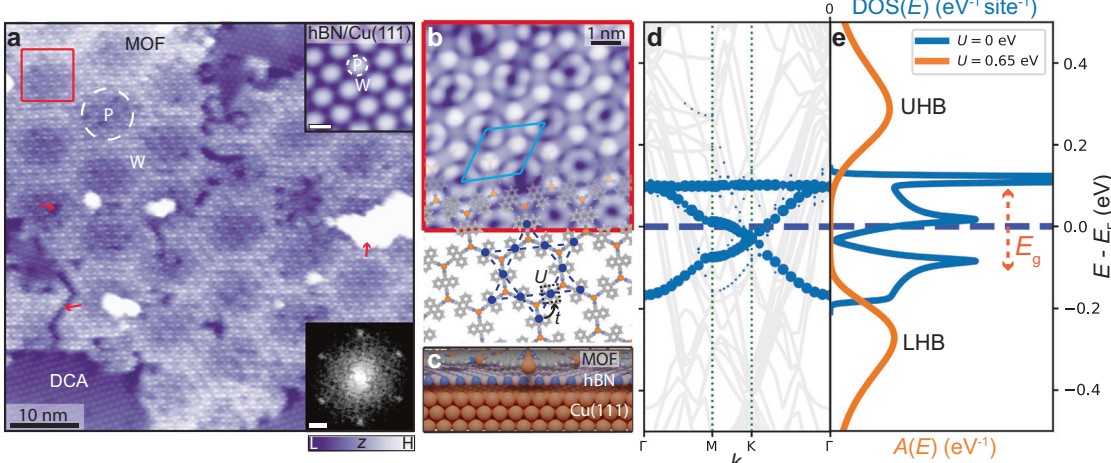

**Fig. 1 | A 2D kagome MOF on an atomically thin insulator: DCA$_3$Cu$_2$ on single-layer hBN on Cu(111). a** STM image of MOF (with organic DCA-only regions; $V_b = -1$ V, $I_t = 10$ pA). 'P' and 'W' indicate pore (dashed white circle) and wire regions of hBN/Cu(111) moiré pattern. Lower inset: Fourier transform of STM image; sharp spots correspond to MOF hexagonal periodicity (scale bar: 0.25 nm$^{-1}$). Upper inset: STM image of bare hBN/Cu(111) moiré pattern ($V_b = 4$ V, $I_t = 100$ pA, scale bar: 4 nm). Red arrows: examples of crack, vacancy, and Cu cluster defects within MOF domain. **b** Top-view of MOF model overlaid upon small-scale STM image of region within red box in **a** ($V_b = -1$ V, $I_t = 10$ pA). MOF unit cell indicated in light blue. Blue dashed lines and solid circles: kagome pattern formed by DCA molecules, with inter-site electron hopping, $t$, and on-site Coulomb repulsion, $U$. **c** Model of MOF/hBN/ Cu(111) (side view). Hydrogen: white; carbon: grey; boron: pink; nitrogen: blue; copper: orange. **d** Electronic band structure calculated by DFT (with $U = 0$)[36]. Blue circles: projections onto MOF states. Grey curves: Cu(111) states (hBN states do not contribute within the shown energy range). Hybridisation between MOF and Cu(111) is hindered by the hBN monolayer; the MOF band structure retains its kagome character. Chemical potential $E_F$ (blue dashed line) is close to half-filling of kagome bands. **e** Density of states, DOS($E$) (tight-binding model with thermal broadening, $U = 0$, blue), and spectral function, $A(E)$ (DMFT, $U = 0.65$ eV, orange), of freestanding MOF. Local Coulomb interaction opens a significant Mott energy gap $E_g$ between lower (LHB) and upper Hubbard bands (UHB).

 

tunnelling spectroscopy (STS); $dI/dV$ is an approximation of the local DOS [$A(E)$] in the non-interacting (interacting, respectively) picture. We performed STS at the ends of the DCA anthracene moiety and at the Cu sites of the MOF—locations where we expect the strongest signature of the kagome bands based on the spatial distribution of the orbitals that give rise to these bands[25,33,34]. These spectra (Fig. 2a), taken at a pore region of the hBN/Cu(111) moiré pattern, both show broad peaks at bias voltages $V_b \approx -0.2$ and $0.2$ V. In a bias voltage window of $\sim0.2$ V around the chemical potential $E_F$ ($V_b = 0$), the $dI/dV$ signal is low, significantly smaller than that for bare hBN/Cu(111).

STM images acquired within this low-$dI/dV$ bias voltage window (Fig. 2b, c) show mainly the topography of the MOF, with the molecules appearing as ellipses of uniform intensity and the Cu atoms as weak protrusions. Outside the low-$dI/dV$ bias voltage window ($|V_b| > 200$ mV), Cu sites and the ends of the DCA anthracene moieties appear as bright protrusions (Fig. 2d, e), similar to the spatial distribution of the electronic orbitals of the DCA$_3$Cu$_2$ MOF associated with the near-Fermi kagome bands (right inset of Fig. 2d; see Supplementary Figs. 11 and 12 for more $V_b$-dependent STM images and $dI/dV$ maps)[25,33,34].

This suggests that the $dI/dV$ peaks at $|V_b| \approx 0.2$ V in Fig. 2a are related to intrinsic MOF electronic states near $E_F$, with the low-$dI/dV$ bias voltage window of $\sim0.2$ V around $E_F$ representing an energy gap, $E_g$, between these states. This is consistent with in-gap topographic STM imaging in Fig. 2b, c[43]. These $dI/dV$ peaks cannot be attributed to inelastic tunnelling (e.g., MOF vibrational modes) as they are not always symmetric about $E_F$ (see Fig. 3). The gap is much larger than that predicted from spin-orbit coupling in such a system[19], and the monocrystalline growth of the MOF domains (Fig. 1a) makes a large disorder-related gap unlikely. Furthermore, the gap is inconsistent with DFT and TB calculations (Fig. 1d, e).

The MOF spectra in Fig. 2a strongly resemble the DMFT-calculated spectral function $A(E)$ in Fig. 1e, including an energy gap of the same magnitude between two similar peaks (Supplementary Fig. 9). This suggests that these $dI/dV$ spectra are hallmarks of a Mott insulator.

## Template-assisted Mott metal-insulator transition (MIT)

We further measured $dI/dV$ spectra at Cu sites of the DCA$_3$Cu$_2$ MOF across the hBN/Cu(111) moiré pattern (Fig. 3a, b). In Fig. 3b, $E_g$ is centred symmetrically about $E_F$ for spectra taken in the middle of a pore region, while those taken closer to the wire region show the Hubbard bands shifting upwards in energy (lowering the barrier to creation of a hole). At the centre of the wire region, the gap at the Fermi level vanishes with a clear increase in Fermi level $dI/dV$ signal (Fig. 3e). The same behaviour was observed for DCA lobe sites of the MOF (see Supplementary Note 14).

The hBN/Cu(111) moiré pattern consists of a modulation of the local work function $\Phi$ (with little structural corrugation), where the quantity $\Delta\Phi = \Phi_{wire} - \Phi_{pore}$ depends on the period of the moiré superstructure, $\lambda$[29–31]. For the hBN/Cu(111) domain in Fig. 3a with $\lambda \approx 12.5$ nm, $\Delta\Phi \approx 0.2$ eV (Fig. 3c; see Supplementary Note 15 for moiré domains with different periods)[30]. Due to energy level alignment[44–46], this corrugation of $\Phi$ affects substrate-to-MOF electron transfer and hence the effective electron filling of the MOF bands, with this filling smaller at wire than pore regions[44,47]. This is consistent with the effective reduction of the hole-creation barrier at the wire relative to the pore in Fig. 3b.

To capture the effect of this moiré-induced modulation of $\Phi$ on the MOF electronic properties, we conducted further DMFT calculations. Using $U = 0.65$ eV (the same as Fig. 1e), we calculated $A(E)$ for a range of $E_F$ assuming a uniform system. We considered a sinusoidal variation of $E_F$ from a minimum value corresponding to half-filling of the kagome MOF bands, with an amplitude of $0.2$ eV, to match the experimental $\Delta\Phi$ for this specific hBN/Cu(111) moiré domain[30,47] (Fig. 3c; see Methods). The obtained $A(E)$ (Fig. 3d) reproduce the experimental spectral features in Fig. 3b, including the shifting of the lower Hubbard band maximum (LHBM) and upper Hubbard band minimum (UHBM) (Fig. 3f), and the vanishing of the gap and increase of the spectral function at the Fermi level for the wire region (Fig. 3e, g).

The DMFT-calculated spectral functions $A(E)$ with no gap at the Fermi level at the smallest electron filling ($\Delta\Phi_{DMFT} = 0.2$ eV) show

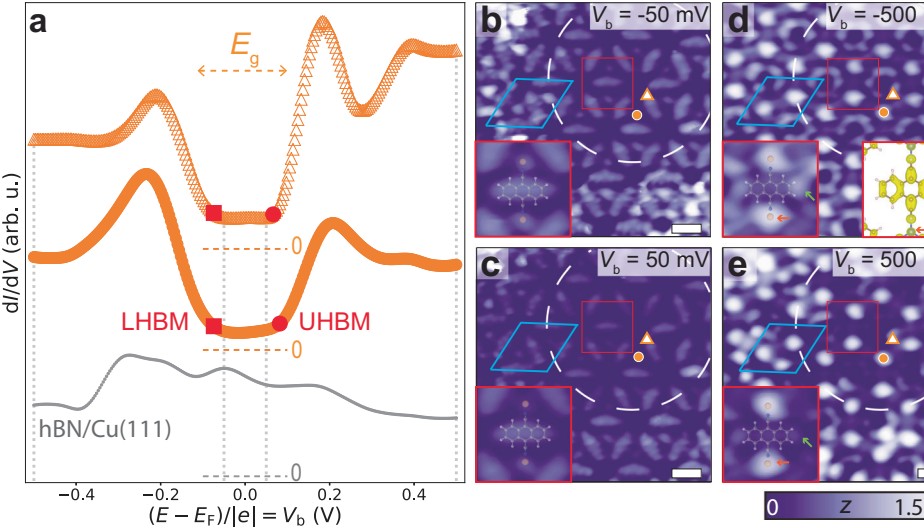

**Fig. 2 | Bandgap $E_g \approx 200$ meV in DCA$_3$Cu$_2$ MOF on hBN/Cu(111). a** $dI/dV$ spectra at MOF positions indicated by orange markers in **b**–**e** and on bare hBN/Cu(111) (grey dots) (setpoint: $V_b = -500$ mV, $I_t = 500$ pA). Grey dotted vertical lines indicate bias voltages at which STM images were acquired in **b**–**e**. Spectra offset for clarity. Dashed horizontal lines indicate $dI/dV = 0$ reference for each curve. Spectra reveal a bandgap $E_g \approx 200$ meV. The non-zero $dI/dV$ within the gap is due to states of underlying Cu(111) leaking through hBN. Red squares (circles): lower Hubbard band maxima, LHBM (upper Hubbard band minima, UHBM). **b**–**e** STM images of MOF on hBN/Cu(111) at specified bias voltages ($I_t = 10$ pA). White dashed circle: hBN/Cu(111) moiré pore. MOF unit cell indicated in light blue. Scale bars: 1 nm. Insets: zoom-in of region within red box, with overlaid Cu-DCA-Cu chemical structure, showing significant contributions from the ends of the DCA anthracene moiety (green arrow) and Cu (orange arrow) for $V_b <$ LHBM and $V_b >$ UHBM. Right inset in **d**: charge density isosurface ($0.0025$ e$^-$ Å$^{-3}$) of DCA$_3$Cu$_2$ obtained by integration of near-Fermi ($\pm0.5$ eV) DFT wavefunctions.

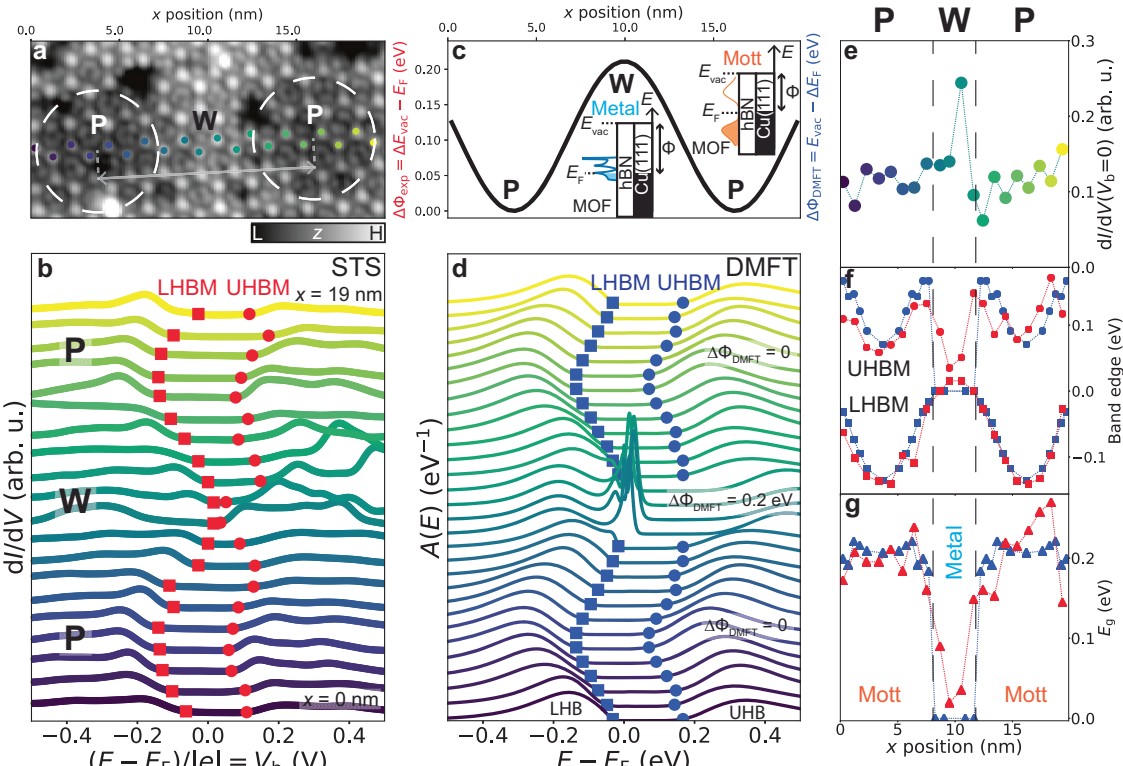

**Fig. 3 | Variation of Mott energy gap in DCA₃Cu₂ MOF induced by hBN/Cu(111) moiré modulation of local work function. a** STM image of MOF ($V_b = -1$ V, $I_t = 10$ pA). White dashed circles (P): hBN/Cu(111) moiré pores, separated by wire (W). Grey arrow indicates moiré period $\lambda \approx 12.5$ nm. **b** d$I$/d$V$ spectra acquired at MOF Cu sites, at positions indicated by coloured markers in **a** (tip 190 pm further from STM setpoint $V_b = 10$ mV, $I_t = 10$ pA). Energy gap $E_g \approx 200$ meV at P regions, vanishing at W region (LHBM: lower Hubbard band maximum; UHBM: upper Hubbard band minimum). **c** Sinusoidal variation of work function, $\Delta\Phi = \Delta E_{vac} - E_F$ ($E_{vac}$: vacuum energy level), across hBN/Cu(111) moiré domain with periodicity

$\lambda \approx 12.5$ nm[30], affecting the MOF electron filling. **d** Spectral functions $A(E)$ calculated via DMFT ($U = 0.65$ eV, $t = 0.05$ eV) for isolated uniform DCA₃Cu₂, for different values of $E_F$. We account for experimental corrugation $\Delta\Phi$ by varying $E_F$ sinusoidally with an amplitude of 0.2 eV as per **c** (see Methods). **e** Experimental d$I$/d$V$ signal at Fermi level ($V_b = 0$) as a function of $x$ position, from **b**. Increased d$I$/d$V(V_b = 0)$ indicates metallic phase. **f, g** Experimental (red; **b**) and DMFT (blue; **d**) LHBM (squares), UHBM (circles), and energy gap $E_g$ (triangles), as a function of $x$ in **a** (experiment) or corresponding $E_F$ (DMFT).

peaks near $E_F$ (Fig. 3d), however, which were not observed in the experimental spectra at the wire region (Fig. 3b). These peaks are indicative of coherent quasiparticles[38], with their width associated with the quasiparticle lifetime and quasiparticle mean free path $\ell$. Via our DMFT and TB calculations (Supplementary Note 3), we estimate $\ell \approx 10$ nm, much larger than the wire region width of $\sim$4 nm. We hypothesise that the coherence peaks are suppressed in the experiment as quasiparticles are strongly scattered by the pore regions (where the MOF remains insulating). This is a key difference between Fig. 3b, d: the experimental measurements represent changes in electron population at finite size MOF regions due to a locally varying work function, whereas each theoretical spectrum corresponds to an infinite uniform system with a constant chemical potential.

Furthermore, some of the experimental spectra in the vicinity of the wire region in Fig. 3b feature narrow peaks at energies of $\sim$0.4 eV, not present in the theoretical $A(E)$ in Fig. 3d. We claim that these peaks do not represent intrinsic electronic states, but are instead related to charging phenomena not captured by the DMFT calculations (see Fig. 4 and Supplementary Note 17).

Despite these discrepancies with experiment, our DMFT calculations capture the fundamental electronic properties of the 2D DCA₃Cu₂ MOF, hosting a Mott insulating phase with $E_g \approx 200$ meV, and a metal-like phase (with no gap at the Fermi level) at the wire region (for the specific tip-sample distance considered in Fig. 3). DMFT is a well-established method for understanding the Mott insulator and Mott MITs[38–41]. This claim of strong electronic correlations and of a Mott insulating phase for the DCA₃Cu₂ kagome MOF is consistent with previous literature[25,27,36,42].

## Tip-assisted Mott MIT at moiré wire region

To explore the nature of the metal-like phase observed at the wire region, we conducted further d$I$/d$V$ STS of the MOF as a function of tip-sample distance $\Delta z + z_0$ (where $z_0$ is set by tunnelling parameters), at a DCA lobe site within the wire region (Fig. 4d). For large $\Delta z + z_0$, these spectra feature an energy gap $E_g$, with a small d$I$/d$V$ signal at the Fermi level (Fig. 4f), similar to spectra in the pore regions (Figs. 2a, 3b; Supplementary Note 20), with a sharp peak at positive $V_b$ (purple circles in Fig. 4d) and a subtler band edge (red squares; similar to band features in Fig. 3b) at negative $V_b$. As $\Delta z$ decreases, the energy position of the sharp peak decreases linearly (Fig. 4e). Conversely, the energy position of the subtler band edge increases with decreasing $\Delta z$, non-linearly and at a lower rate (Fig. 4e). These features cross the Fermi level at intermediate $\Delta z$, and the spectrum becomes gapless ($E_g = 0$), with a significant increase in d$I$/d$V$ signal at the Fermi level (Fig. 4f). Note that an intermediate $\Delta z$ was also used for all spectra in Fig. 3b where similar metal-like signatures were observed at the wire region. We found a similar $\Delta z$-dependent trend for wire region Cu sites (Supplementary Note 21). Notably, moiré pore regions remain gapped for all $\Delta z$ values (for both Cu and DCA lobe sites; Supplementary Note 20 and Supplementary Note 21).

As $V_b$ is applied between the tip and Cu substrate, the STM double-barrier tunnel junction (DBTJ)—where the vacuum between tip and MOF is a first tunnel barrier and the insulating hBN is a second one—causes a voltage drop at the MOF location. This can lead to energy shifts of MOF states and/or charging of such states when they become resonant with the Cu(111) Fermi level (Fig. 4a)[33,46,48]. These phenomena

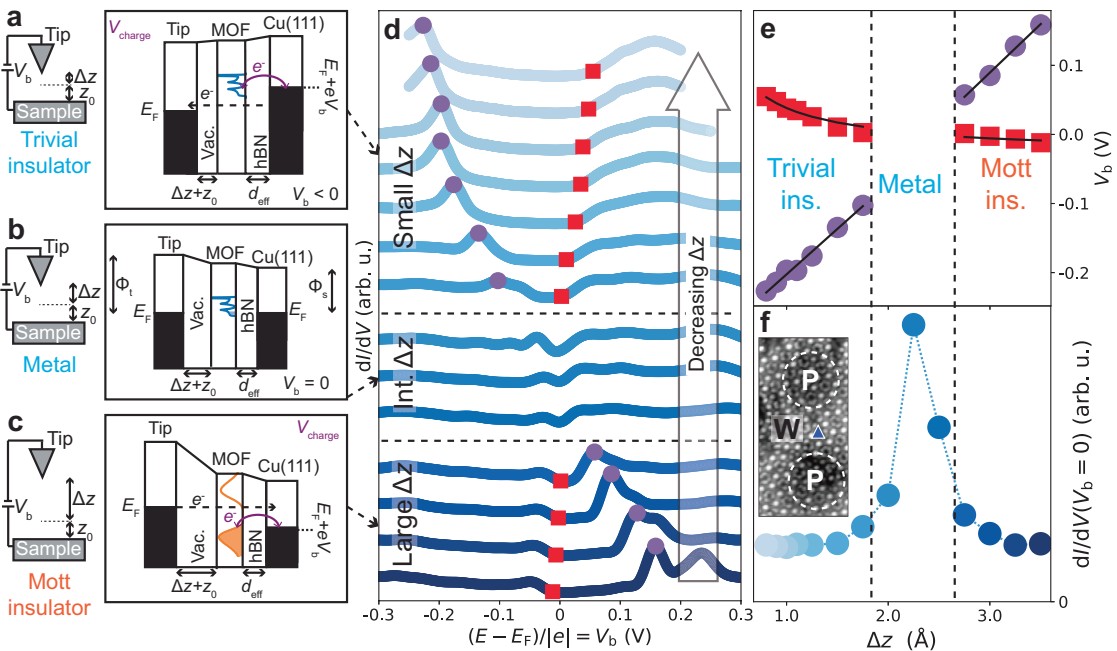

**Fig. 4 | Mott metal-insulator transition controlled via tip-induced gating.**
**a–c** Schematics and energy diagrams of tunnelling and charging processes at a
double-barrier tunnel junction (DBTJ), consisting of STM tip, vacuum barrier (vac.),
MOF, hBN barrier, and Cu(111), for small, intermediate (int.), and large tip-sample
distances ($\Delta z + z_0$). When $V_b$ is applied, voltage drop at MOF location enables tip-
controlled charging, energy level shifts, and gating of MOF transitions, from (cor-
related) Mott insulator to metal to trivial insulator. These schematics qualitatively
illustrate our proposed physical mechanism of tip-induced gating. **d** d$I$/d$V$ spectra
at MOF DCA lobe site, at hBN/Cu(111) moiré wire, for different $\Delta z + z_0$ ($z_0$ given by

STM setpoint $V_b = 10$ mV, $I_t = 10$ pA). Purple circles: MOF charging peak. Red
squares: intrinsic electronic state at MOF band edge. Spectra normalised and offset
for clarity. **e** $V_{charge}$ (purple circles in **d**) and $V_{state}$ (red squares in **d**) as a function of
$\Delta z$. Black solid lines: global fits to Eqs. (1) and (2). **f** d$I$/d$V$ signal at Fermi level ($V_b = 0$)
as a function of $\Delta z$, from **d**. Increased d$I$/d$V(V_b = 0)$ indicates metallic phase at
intermediate $\Delta z + z_0$. Inset: STM image of MOF on hBN/Cu(111) showing site (blue
triangle marker) where d$I$/d$V(\Delta z)$ measurements were performed ($V_b = -1$ V,
$I_t = 10$ pA, scale bar: 2 nm).

have previously been observed for the DCA$_3$Cu$_2$ MOF on a decoupling
graphene surface[33]. In this scenario, the bias voltages corresponding to
an intrinsic electronic state, $V_{state}$, and to charging of such a state,
$V_{charge}$, vary as a function of $\Delta z$ as:

$$V_{state}(\Delta z) = \frac{d_{eff}(V_\infty + \Delta\Phi_{ts})}{(\Delta z + z_0)} + V_\infty, \qquad (1)$$

$$V_{charge}(\Delta z) = -\frac{V_\infty(\Delta z + z_0)}{d_{eff}} - (V_\infty + \Delta\Phi_{ts}), \qquad (2)$$

where $d_{eff}$ is the effective width of the hBN tunnel barrier, $V_\infty$ is the bias
voltage corresponding to the electronic state as $\Delta z \to \infty$, and $\Delta\Phi_{ts}$ is the
difference between tip and sample work functions[48].

We fit the $\Delta z$-dependent bias voltage associated with the subtle
band edge (red squares) in Fig. 4d with Eq. (1), and the bias voltage
associated with the sharp peak (purple circles) with Eq. (2) (Fig. 4e).
The agreement between experimental data and fits indicate that the
subtle spectral band edge (red) represents an intrinsic MOF electronic
state, with its energy shifting as $\Delta z$ varies, and with the sharp peak
(purple) corresponding to charging of such a state.

## Discussion

We interpret these results as follows. For large $\Delta z$ (Fig. 4c), the MOF
electronic states are strongly pinned to the substrate, and the MOF
near-Fermi kagome bands are approximately half-filled. Here, the d$I$/
d$V$ spectra feature an energy gap at the Fermi level, both at moiré pore
(Figs. 2a, 3b; Supplementary Note 20 and Supplementary Note 21) and
wire regions (bottom spectra of Fig. 4d; Supplementary Note 21): the
entire monolayer 2D MOF is intrinsically a Mott insulator featuring
localised electrons. This is consistent with DMFT calculations, which

show that the system can remain Mott insulating even with a 0.2 eV
modulation of $E_F$ (corresponding to the experimental moiré work
function corrugation; Supplementary Fig. 4).

Due to the hBN/Cu(111) moiré work function corrugation, the
LHBM at a wire region is very close to the Fermi level (in comparison to
the pore region). As $\Delta z$ is reduced, the MOF states become less pinned
to the Cu(111) substrate and more pinned to the tip via the DBTJ effect
(Fig. 4; Supplementary Note 19, Supplementary Note 20, Supplemen-
tary Note 21). Given that $\Phi_{tip} > \Phi_{wire}$ (see Supplementary Note 18), this
leads to an energy upshift of the MOF states with respect to the Cu(111)
Fermi level (with the LHBM susceptible to charging as $V_b$ becomes
more positive). At intermediate $\Delta z$, this energy upshift depopulates the
LHB and leads to the transition from the Mott insulating phase (only
existing at half-filling of the three near-Fermi kagome bands; Fig. 1) to
the metallic phase. This is concomitant with a dramatic change in the
d$I$/d$V$ spectra, including the vanishing of the energy gap and the
increase in d$I$/d$V$ signal at the Fermi level. As $\Delta z$ is further reduced
(Fig. 4a), the near-Fermi kagome bands become fully depopulated
(with the bottom of these bands susceptible to charging as $V_b$ becomes
more negative; top spectra of Fig. 4d), and the MOF becomes a trivial
insulator (with a gap between these near-Fermi bands and lower
energy bands; Supplementary Fig. 1). The DBTJ effect also manifests
itself at the pore region, with energy shifts of the LHBM and UHBM as
$\Delta z$ varies (Supplementary Note 20, Supplementary Note 21). Yet, due
to the smaller $\Phi_{pore}$ (Fig. 3c), the Fermi level lies close to the centre of
the energy gap (Fig. 3b), making the Mott insulating phase robust for
the considered $\Delta z$ range, consistent with DMFT (Fig. 3d, Supplemen-
tary Figs. 2b, 4). In the wire region, the STM tip, via the DBTJ effect, in
combination with the large $\Phi_{wire}$, acts as a local electrostatic gate,
switching the 2D MOF from Mott insulator to metal (Supplemen-
tary Fig. 26).

This tip-induced gating at the wire region is inherently local, occurring within the cross-section of the DBTJ (typically with a diameter of ~10 nm given by the tip radius of curvature). This locality could lead to $dI/dV$ features (e.g., due to electronic confinement) not captured by DMFT. Whether these local changes produce truly delocalised metallic states across an extended area of the sample remains an open question, beyond the scope of this investigation. Future work could investigate the $DCA_3Cu_2$ MOF on single-crystal exfoliated hBN, within a gated heterostructure, allowing for uniform electrostatic control and bulk (e.g., transport) measurements.

Our assertion that the MOF is a Mott insulator at the moiré pore regions for all tip-sample distances, and at the wire regions for large tip-sample distances, is well supported by DMFT calculations, and by previous literature[25,27,36,42]. The DMFT spectral functions demonstrate excellent agreement with $dI/dV$ spectra at the pore regions, including the ~200 meV gap at the Fermi level, and the energy modulation of the LHB and UHB due to variations in electrostatic potential. Our assertion of a metallic phase for the MOF at the wire region for intermediate tip-sample distances is well supported by the DBTJ model, and by the qualitative agreement with DMFT, including the absence of an energy gap at the Fermi level and an increase in Fermi level $dI/dV$ signal and spectral function.

Mott insulating—with intrinsically localised electronic states—and metallic phases have been observed in $TaS_2$ and $TaSe_2$ monolayers, which feature frustrated lattice geometries and lattice constants (due to charge density wave distortions) similar to our work[3,4,49–51], for finite domain sizes as small as ~$10 \times 10$ nm$^2$ [49]. Also, local topological phase transitions induced by an STM tip have been demonstrated[52]. Our interpretation of a local population-induced Mott metal-insulator transition is consistent with these findings.

Monolayer DCA-based MOFs have been studied on other substrates[25,26,33–35,37,53–55], without observing a Mott phase. In our case, the combination of the wide bandgap hBN as a template (allowing the MOF to retain its intrinsic electronic properties), and of the adequate energy level alignment given by the hBN/Cu(111) substrate (resulting in half-filling of kagome bands; Fig. 1d), plays a key role in the realisation of the correlated-electron Mott phase.

Note that the vanishing of the energy gap at the Fermi level for the MOF metallic phase at the moiré wire region (Fig. 3b) can be reminiscent of spectral features for the pseudogap phase in cuprates and other transition metal oxides[56–58]. Future work on controllable Mott MITs in MOFs might shed light on the general understanding of doped Mott insulators.

We have demonstrated that single-layer $DCA_3Cu_2$ not only hosts a robust Mott insulating phase (with $E_g \gg k_B T$ at $T = 300$ K), but also that Mott MITs can be achieved via the combination of template- (Fig. 3) and tip- (Fig. 4) induced gating, consistent with DMFT and the DBTJ model. This shows that such phase transitions can be controlled in monolayer MOFs via electrostatic tuning of the chemical potential.

Our findings represent a promising step towards incorporation of 2D MOFs as active materials in device-like architectures (e.g., van der Waals heterostructures based on 2D materials), benefiting from efficient synthesis approaches and versatility offered by MOFs, and allowing for access and control of correlated-electron phases therein via electrostatic gating[59]. Our work establishes single-layer 2D MOFs—with crystal geometries allowing for flat bands—as promising platforms for controllable switching between diverse many-body quantum phenomena, potentially including correlated magnetism, superconductivity, and quantum spin liquids.

## Methods
### Sample preparation
The monolayer (ML) $DCA_3Cu_2$ kagome MOF was synthesised on hBN/Cu(111) in UHV (base pressure ~$2 \times 10^{-10}$ mbar). The Cu(111) surface was first cleaned via 2–3 cycles of sputtering with $Ar^+$ ions and subsequent annealing at ~770 K. A hBN ML was synthesised on Cu(111) via the thermal decomposition of borazine[31]. We dosed a partial pressure of borazine of ~$9 \times 10^{-7}$ mbar for 45 minutes with the Cu(111) sample maintained at 1140 K. We kept the Cu(111) sample at this temperature for a further 20 mins to ensure a complete reaction. We then cooled the sample to room temperature and deposited the DCA molecules via sublimation at 390 K, corresponding to a deposition rate of 0.007 ML/sec. In our experiments we considered DCA coverages of ~0.4–0.6 ML. We then further cooled the sample to ~77 K before depositing Cu via sublimation at 1250 K (Cu deposition rate: ~0.002 ML/sec; typical Cu coverages in our experiments: ~0.05 ML). Finally, the sample was annealed to ~200 K for 15 minutes. Further details are in Supplementary Note 6.

The $DCA_3Cu_2$ MOF crystalline structure was found to be commensurate with the hBN lattice but incommensurate with the long-range hBN/Cu(111) moiré patterns of different sizes, across which the MOF grows without disruption (see Supplementary Fig. 8). The hBN/Cu(111) moiré pattern is clearly visible in large-scale STM images of the MOF (Fig. 1a). This is consistent with the modulation of the MOF's electronic properties illustrated in Fig. 3 (also see $dI/dV$ maps in Supplementary Fig. 12).

### STM and STS measurements
All STM and $dI/dV$ STS measurements were performed at 4.5 K (except measurements in Supplementary Note 23 performed at 77 K), at a base pressure <$1 \times 10^{-10}$ mbar, with a hand-cut Pt/Ir tip. All STM images were acquired in constant-current mode with tunnelling parameters as reported in the text (bias voltage applied to sample). All $dI/dV$ spectra were obtained by acquiring $I(V)$ at a constant tip-sample distance (stabilised by a specified setpoint tunnelling current and bias voltage), and by then numerically differentiating $I(V)$ to obtain $dI/dV$ as a function of bias voltage. Tips were characterised on regions of bare hBN/Cu(111) prior to spectroscopy measurements, where the Shockley surface state of Cu(111) could be observed (grey curve in Fig. 2a, onset shifted upwards in energy due to confinement by hBN monolayer)[29].

### DFT calculations
We calculated the non-spin-polarised band structure of $DCA_3Cu_2$ on hBN on Cu(111) via DFT (Fig. 1d), using the Vienna Ab-Initio Simulation Package[60] with the Perdew-Burke-Ernzerhof functional under the generalised gradient approximation[61]. We used projector augmented wave pseudopotentials[62,63] to describe core electrons, and the semi-empirical potential DFT-D3[64] to describe van der Waals forces.

The substrate was modelled as a slab three Cu atoms thick, with the bottom layer fixed at the bulk lattice constant[65]. A layer of passivating hydrogen atoms was applied to the bottom face to terminate dangling bonds.

A 400 eV cut-off was used for the plane wave basis set. The threshold for energy convergence was $10^{-4}$ eV. The atomic positions of the $DCA_3Cu_2$/hBN/Cu(111) were relaxed until Hellmann-Feynman forces were <0.01 eV/Å, using a $3 \times 3 \times 1$ k-point grid for sampling the Brillouin zone and 1st order Methfessel-Paxton smearing of 0.2 eV. The charge density for the relaxed structure was calculated using an $11 \times 11 \times 1$ k-point grid, Blöchl tetrahedron interpolation, and dipole corrections. The band structure was determined non-self-consistently from the charge density.

Note that small (<1 Å) perturbations in the height of $DCA_3Cu_2$ above the hBN/Cu(111) do not appreciably affect the calculated band structure. As such, small perturbations in height related to the hBN/Cu(111) moiré pattern (of at most 0.7 Å[66]) were not captured by these calculations[36].

### DMFT calculations
We performed DMFT calculations on the freestanding $DCA_3Cu_2$ kagome MOF. We used the Hubbard model for a kagome lattice with

nearest-neighbour hopping,

$$H = -t \sum_{\langle i,j \rangle, \sigma} \hat{c}^{\dagger}_{i,\sigma} \hat{c}_{j,\sigma} + U \sum_i \hat{n}_{i,\uparrow} \hat{n}_{i,\downarrow}, \qquad (3)$$

where the first term is the TB Hamiltonian with nearest-neighbour hopping energy $t$, $\sum_{\langle i,j \rangle}$ is a sum over nearest-neighbour sites, and the second term is the interaction Hamiltonian with on-site Coulomb repulsion $U$. The operator $\hat{c}^{\dagger}_{i,\sigma}$ ($\hat{c}_{i,\sigma}$) creates (annihilates) an electron at site $i$ with spin $\sigma$; $\hat{n}_{i,\sigma} = \hat{c}^{\dagger}_{i,\sigma} \hat{c}_{i,\sigma}$ is the density operator. We take $t = 0.05$ eV to match prior DFT calculations of $DCA_3Cu_2$[19,27,36].

We first calculated the non-interacting ($U = 0$) density of states (DOS; blue curve in Fig. 1e) by numerically integrating over all momenta in the first Brillouin zone. The chemical potential $E_F$ in Fig. 1e was chosen to be consistent with the electron filling predicted by DFT (Fig. 1d). We applied a thermal broadening ($k_BT = 2.5$ meV) to this non-interacting TB DOS to make it consistent with the thermal broadening of the DMFT-generated (see below) spectral function $A(E)$ in Fig. 1e.

To account for electronic correlations, we then implemented the DMFT formalism[38,67,68] using the Toolbox for Researching Interacting Quantum Systems (TRIQS)[69], with the continuous-time hybridisation expansion solver (CTHYB)[70,71] at a temperature of ~29 K ($k_BT \approx 0.05t$, unless specified otherwise; see Supplementary Note 23 for temperature-dependent calculations), with $U = 0.65$ eV. To use a single-site DMFT formalism[38] with the kagome band structure, the non-interacting DOS of the three kagome bands were combined into a single function for use as the input into the DMFT procedure.

We calculated the many-body spectral functions $A(E)$ (analogous to the DOS, but in the interacting regime; Fig. 3d) via analytic continuation using the maximum entropy method (MaxEnt) as implemented[72] in TRIQS. The meta-parameter, $\alpha$, was determined from the maximum curvature of the distance between the MaxEnt fit and data, $\chi^2$, as a function of $\alpha$[73].

Each DMFT calculation assumed a spatially uniform work function $\Phi$; long-range modulation of $\Phi$ is beyond the capabilities of DMFT. As such, the spatially varying sample work function $\Phi$ resulting from the experimental hBN/Cu(111) moiré pattern was not explicitly captured in the individual $A(E)$ spectra in Fig. 3d. This spatial variation of $\Phi$ was approximated by varying the uniform $E_F$ of the system for each individual $A(E)$ spectrum. Each of these $A(E)$ spectra was then associated to a specific location of the hBN/Cu(111) moiré pattern (and hence to a specific experimental dI/dV curve) based on how this variation of $E_F$ would translate to a local $\Phi$. These calculations assume that the theoretical $A(E)$ spectra, calculated with a uniform $E_F$ (and hence uniform $\Phi$), are reasonable representations of the locally acquired experimental dI/dV curves, which are affected by a spatially varying $\Phi$. This assumption is reasonable for the insulating phase. Indeed, in the Mott insulating phase, electronic states are localised at the kagome sites, confined within areas that are small in length[74] compared to the distance between nearest-neighbour kagome sites (~1 nm) and to the periodicity $\lambda$ of the hBN/Cu(111) moiré domains considered in our experiments ($\lambda > 5$ nm). DMFT indicates that this Mott insulating phase is robust to variations in chemical potential $E_F$ larger than 0.2 eV, with the spectral function $A(E)$ shifting in energy as $E_F$ is varied within this range, without other significant qualitative changes (see Supplementary Fig. 2b for $E_F = 0.25$ to 0.5 eV). In our experiments, the hBN/Cu(111) moiré pattern imposes a periodic modulation of the local work function $\Phi$, with a peak-to-peak modulation amplitude of ~0.2 eV and a modulation periodicity $\lambda \approx 12.5$ nm (Fig. 3c; this amplitude becomes smaller with decreasing $\lambda$, see Supplementary Fig. 22). That is, $\Phi$ varies *slowly* across the MOF kagome lattice. The effect of such long-range modulation of $\Phi$ is to shift the energy of the MOF localised states accordingly. As long as the $\Phi$ modulation period is larger than the distance between nearest-neighbour kagome sites and the $\Phi$ modulation amplitude is smaller than a critical value inducing the transition

to the metallic phase, there is no other dramatic qualitative effect on these localised electronic states. This explains the excellent agreement between experimental dI/dV spectra for the MOF at the hBN/Cu(111) moiré pore region and DMFT-calculated spectral functions $A(E)$ for the system in the Mott insulating phase (Fig. 3b, d), with LHB and UHB modulated in energy following the variation in electrostatic potential. For the MOF in the metallic phase at the moiré wire region (Figs. 3, 4), discrepancies between theory and experiment can be explained (as discussed in the main text) by long electronic coherence lengths (Supplementary Note 3), and by effects of long-range moiré $\Phi$ modulation and of finite DBTJ cross section on the potentially delocalised metallic MOF states.

### DBTJ model

In Fig. 4d, e, the bias voltage, $V_{state}$, corresponds to the energy level of an intrinsic MOF frontier electronic state (which is susceptible to charging), at a band edge in either the Mott or trivial insulator regime. Red square markers indicating $V_{state}$ were assigned based on the method outlined in Supplementary Note 13. The bias voltage, $V_{charge}$, corresponds to the peak associated with charging of such a MOF state. Purple circle markers indicating $V_{charge}$ were assigned by finding a local maximum in dI/dV.

In Fig. 4e, we considered four experimental datasets for fitting with Eqs. (1) and (2): $V_{state}(\Delta z)$ (red squares) and $V_{charge}(\Delta z)$ (purple circles) for small values of $\Delta z$ (trivial insulator phase), and $V_{state}(\Delta z)$ and $V_{charge}(\Delta z)$ for large values of $\Delta z$ (Mott insulator phase). Given the Mott MIT, we considered two different intrinsic MOF band edges susceptible to charging, embodied in two different values of $V_\infty$: one for the trivial insulator phase (small values of $\Delta z$) and one for the Mott insulator phase (large $\Delta z$). This phase transition is evident from the offset in $V_{state}$ (red squares in Fig. 4e) observed when $\Delta z$ varies from small to large (through the metallic phase at intermediate $\Delta z$). Accordingly, we used a global fitting approach to obtain the same fitting parameters $d_{eff}$, $z_0$, and $\Delta\Phi_{ts}$ (characteristic of the DBTJ and the acquisition location) for these four experimental datasets, and a separate $V_\infty$ value for each regime.

Schematics in Fig. 4a–c represent cartoon illustrations of our proposed physical mechanism for tip-induced gating.

It is important to note that the DBTJ inherently affects all dI/dV measurements in this work, including those in Figs. 2 and 3, for both pore and wire regions of the hBN/Cu(111) moiré pattern. The DBTJ effect does not cause phase transitions at the pore regions, however (see Supplementary Note 20, Supplementary Note 21). The measurements in Fig. 3 were performed with an intermediate tip-sample distance—which is why metallic properties were observed at the wire region.

## Data availability

The data supporting the findings of this study is available from the authors upon request.

## Code availability

All codes relating to DMFT and MaxEnt are available at https://doi.org/10.5281/zenodo.7439858. The authors can provide code for data analysis and theoretical calculations upon request.

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

## Acknowledgements

A.S. acknowledges funding support from the ARC Future Fellowship scheme (FT150100426). B.J.P. acknowledges funding support from the ARC Discovery Project scheme (DP180101483). H.L.N. acknowledges funding support from the MEXT Quantum Leap Flagship Programme (JPMXS0118069605). B.L., J.H., J.C., and N.V.M. acknowledge funding support from the Australian Research Council (ARC) Centre of Excellence in Future Low-Energy Electronics Technologies (CE170100039). B.L., B.F., and J.C. are supported through Australian Government Research Training Programme (RTP) Scholarships. B.F. and N.V.M. gratefully acknowledge the computational support from the National Computing Infrastructure and Pawsey Supercomputing Facility. The authors also thank Prof. Michael S. Fuhrer, Prof. Jaime Merino Troncoso, and Dr. Daniel Moreno Cerrada for their valuable discussions.

## Author contributions

B.L., J.H., and A.S. conceived and designed the experiments. B.L., J.H., and J.C. performed the experiments. B.F. and H.L.N. performed the theoretical calculations with guidance from B.J.P. and N.V.M. All authors contributed to writing the manuscript.

## Competing interests

The authors declare no competing interests.
