## [Peer Review File · Nature Communications]

REVIEWER COMMENTS

Reviewer #1 (Remarks to the Author):

B. Lowe et al. report on the DCA₃Cu₂ 2D MOF fabrication (following the concepts of supramolecular coordination chemistry) on h-BN/Cu(111) under ultra-high vacuum (UHV). Its structural and electronic properties are studied by low-temperature scanning tunneling microscopy (STM) and spectroscopy (STS), supported with density functional theory (DFT) and dynamical mean-field theory (DMFT) calculations.

The combination of the wide bandgap h-BN as a template (allowing the 2D-MOF to retain its intrinsic electronic properties), and of the adequate energy level alignment given by the h-BN/Cu(111) substrate (resulting in half-filling of the 2D-MOF kagome bands) promotes the realization of a correlated-electron Mott phase. With STS measurements, they find an electronic energy gap of around 200meV, corresponding to a Mott insulating phase according to DMFT predictions. In addition, by tuning the electron population of the 2D-MOF near-Fermi band structure, via either template induced (work function variation of the pores and wires of the h-BN/Cu(111) electronic moiré) or tip-induced gating (via STM probe sample distance) a Mott metal-insulator transition in the 2D-MOF is proposed.

Even though this work claims the observation of an exotic many-body quantum phenomenon (Mott metal-insulator transition) in a 2D-MOF, the experimental evidence and theoretical support are not strong enough. Therefore, the status of this work is still too preliminary so that it can be published in Nature Communications. The Mott metal-insulator transition should be present in the entire 2D-MOF material (since this effect alters the band structure of the material). Therefore, the present claim that this effect can happen at the local scale (< 5 nm) sounds controversial. Additional experimental evidence, such as temperature dependence studies of the insulating gap size and angle-resolved photoemission spectroscopy measurements are necessary. In addition, the authors should carefully address the following major concerns that I have (points i to xv).

i) The DCA₃Cu₂ 2D-MOF has already been studied on a decoupling layer such as graphene (ref. 31). There a long-range ordered 2D-MOF phase was achieved, by far exceeding the quality of the DCA₃Cu₂ 2D-MOF presented in this manuscript. Even though the authors claim to have achieved a highly crystalline growth, this is not the case so far. The authors must provide better experimental proof to defend this point.

ii) The authors claim in the abstract that correlated-electron phases, or 2D-MOF “quantum materials” have not been experimentally realized yet. However, in a recent paper, already uploaded into arXiv before this manuscript, Lobo-Checa et al. report 2D magnetism in a very similar 2D-MOF i.e., the DCA₃Fe₂ 2D-MOF grown on Au(111) (see <https://arxiv.org/abs/2209.14994>). The authors should give credit to other colleagues working in the same research field. This preprint should be mentioned in the introduction and cited.

iii) The synthesis of a 2D-MOF on h-BN via supramolecular coordination chemistry is not new (see ref.30). In addition, the gate control of the DCA₃Cu₂ 2D-MOF electronic structures and charging was also already performed on graphene (ref. 31).

iv) page 3 (lines 84-85) “a crystalline MOF domain” growth is claimed. However, compared to ref. 31, this is far from crystalline. As shown in Fig. 1(a) of the manuscript, the MOF domain is surrounded by excess DCA molecules, therefore the 3:2 stoichiometry was only successfully achieved on rather small regions of the sample (50 × 50 nm²). In addition, white islands embedded into the 2D-MOF domains may point out to excess Cu islands. Can the authors identify such islands? There are also many additional defects in the MOF domain which are not discussed (black or dark blue points, wires or regions inside the domain). It is therefore almost impossible to find more than two h-BN pore regions together, covered by defect free DCA₃Cu₂ 2D-MOF. The authors should provide experimental evidence of a long-range ordered growth of the 2D MOF. Otherwise, the authors should discuss the defects appearing in the 2D-MOF domains, as well as their limitations to achieve a highly crystalline 2D-MOF growth. This is an important point since such small domain sizes and high defect density will preclude the use of such 2D-MOFs in electronic devices.

v) page 4 (line 87). Here the growth of the 2D-MOF on top of the h-BN/Cu(111) electronic moiré is presented. However, it is known that the moiré periodicity can change approximately from 3 nm to 14 nm on the h-BN/Cu(111) samples grown with borazine in UHV (ref. 30). Is there any moiré periodicity preference for the successful growth of 2D-MOF islands on top? The authors should provide a discussion in this regard.

vi) page 4 (lines 102-105): The authors should be transparent and mention that DCA₃Cu₂ 2D-MOF (the very same systems as the one shown here) has already been grown (as a full monolayer) on top of a rather good decoupling layer of graphene on Ir(111) (ref. 31).

vii) With respect to Figure 1. The authors only focus on the kagome band structure close to the Fermi level. However, it is known that more kagome bands should appear above and below the energy range considered here (ref. 19). Can the authors comment on this point? Do the authors have DFT calculations for a wider energy range? How does the band structure of the Mott insulating phase look like?

viii) The authors should provide STS performed on a larger voltage range (for instance from -2V to +2V) to see what happens to the additional kagome bands away from the Fermi level. Is there any strong molecular HOMO/LUMO or 2D-MOF valence/conduction band feature at lower/higher voltages, as already observed in ref. 33 and in A. Kumar et al. Nano Letters 18, 5596 (2018). The latter reference should be added since DCA₃Co₂ 2D-MOF was grown on the rather good decoupling graphene layer on Ir(111), whereby a 2D-MOF band structure signature was already observed.

ix) page 6 (line 149) “the high crystalline growth makes a large disorder-related gap unlikely”. If this is the case, how does the STS look at different points of the 2D-MOF island? Are the features identical if one performs STS on the dark pore regions? How do the electronic properties look like at the edge of the 2D-MOF domain? At which position does the domain edge already play a role in disturbing the reported electronic properties?

x) With respect to Figure 2. Why are the spectra asymmetric? Why not show dI/dV maps at particular energies instead of STM images?

xi) page 6 (lines 157-162): Figure 3 should be explained in more detail in the manuscript, specially panels 3e and 3f. The order of the figure description should be changed in the text (a, b, c....). Regarding panel 3b, apart from the theoretically predicted peak at the Fermi level, which can not be captured experimentally, there are many additional peaks in the unoccupied region (>0.2 V). These peaks or modulations are not present in the theoretical simulations. What are they? They have a similar amplitude to the relevant peaks close to the Fermi level, therefore, most probably are not artefacts. The authors should describe them.

xii) Figure 3a,b. The authors perform STS at Cu adatom positions. For completeness, the authors should also perform the same STS sequence at the DCA anthracene lobe positions. Does the same Mott metal-insulator transition appear or evolve in the same fashion as on the Cu adatoms?

xiii) Figure 3. Since the Mott metal-insulator transition is highly dependent on the work function variation of the h-BN/Cu(111) electronic moiré, the authors should provide more experimental evidence that this effect can be reproduced for different moiré periodicities.

xiv) In Figure 4 the STS were performed at a DCA anthracene lobe position. This point is not mentioned in the manuscript and is important. For completeness, the authors should perform the gating experiment on a Cu adatom position.

xv) Figure S10. It is not clear at all that the charging rings are increasing their perimeter around the DCA molecule (see ref. 31). It looks just the same intensity at the Cu positions. This is not consistent with the molecular charging ring features reported in ref. 31. These dI/dV maps do not support the interpretation of charging peaks given in Figure 4 of the manuscript.

Reviewer #2 (Remarks to the Author):

In their manuscript entitled "Gate control of Mott metal-insulator transition in a 2D metal-organic framework", B. Lowe et al claim that in a metal-organic network grown on an hBN/Cu(111) substrate it is possible to control a metal-insulator transition through two mechanisms. One of them is based in the modulation of the surface potential due to the presence of a moiré pattern between hBN and Cu(111). In the second part of the manuscript, they control the transition by varying the sample tip distance. There are different aspects of the article that need to be clarified before the article can be published.

Figure 3 shows the spectra measured by moving the tip between the areas of the moiré called pores to the areas called wires. The experimental spectra show a modulation in the position of the bands depending on the moiré area. Upon reaching the areas called wires, the gap disappears and very

intense peaks extend from the Fermi level up to +0.4eV. These spectra features are not mentioned or discussed in the manuscript.

The calculations reproduce the energy position of the bands observed in the experiment until the wire areas where the disagreement with the experiments it is clear. The calculations corresponding to the wire areas show very pronounced and very narrow peaks. The authors attributed their origin to coherent quasiparticle. Contrary to what the authors say in the manuscript, the calculations do not show a gap collapse in the wire areas, on the contrary it becomes wider and extends to almost +0.4eV.

According to the authors, the control of the metal-semiconductor transition induced by the substrate is deduced from the comparison between the experimental results and the theoretical calculations shown in Figure 3. Before this can be stated it is necessary to discuss in detail the discrepancies mentioned above between theory and experiments.

In the second part, the authors show how the metal-insulator transition can be induced by changing the tip-sample distance. To do this, they carry out experiments in one of the areas called wire. Figure 4d shows how in this area the measured spectra may or may not show a gap depending on the tip-sample distance. This result somehow invalidates the manuscript's first claim that moiré registration controls the existence of a gap or not. It seems that the gap collapse depends on the parameters used to perform the measurements.

After reading the manuscript and the supplementary material, it is not clear to me how the assignments of the purple circles and red squares are made in Figure 4 or how it is decided whether the observed gap corresponds to a trivial insulator or a Mott insulator.

No details are given as to whether the energy levels shown in Figure S12 are a cartoon or the result of a calculation. This detail is important to understand how the authors identify the gap character.

Reviewer #3 (Remarks to the Author):

In this work, the authors reported the experimental synthesis and characterization of a single-layer 2D DCA₃Cu₂ MOF on a wide bandgap BN substrate, which is further shown to host a robust Mott insulating phase and can achieve a Mott metal-insulator transition using electrostatic control. The experimental observations are quantitatively consistent with theoretical predictions (both DMFT calculations and DBTJ model). Direct experimental measurements on a Mott metal-insulator transition in 2D MOFs remain elusive. One of the challenges lies in the experimental synthesis of large single-crystal MOF samples. It is nice to see that the authors have succeeded in making one such sample and successfully demonstrated the Mott metal-insulator transitions induced via either template or tip.

However, it is important for the authors to address a question that arises from the examination of the large-scale samples, as depicted in Figures 1a and S4. One can clearly see structural defects, such as vacancies and grain boundaries. As these defects may have a profound influence on the electronic properties of the MOF, it would be good for the authors to have some discussion about the potential impact of these bulk defects. This work represents a noteworthy contribution to the research field of 2D MOFs, shedding light on elucidating the Mott metal-insulator transition in MOFs. I would recommend its publication in Nature Communications after the authors address the aforementioned concerns.

We thank all three reviewers for their comments, which have allowed us to improve our manuscript substantially. Below we have addressed general remarks and point-by-point comments from each reviewer.

Reviewer 1

B. Lowe et al. report on the DCA₃Cu₂ 2D MOF fabrication (following the concepts of supramolecular coordination chemistry) on h-BN/Cu(111) under ultra-high vacuum (UHV). Its structural and electronic properties are studied by low-temperature scanning tunneling microscopy (STM) and spectroscopy (STS), supported with density functional theory (DFT) and dynamical mean-field theory (DMFT) calculations.

The combination of the wide bandgap h-BN as a template (allowing the 2D-MOF to retain its intrinsic electronic properties), and of the adequate energy level alignment given by the h-BN/Cu(111) substrate (resulting in half-filling of the 2D-MOF kagome bands) promotes the realization of a correlated-electron Mott phase. With STS measurements, they find an electronic energy gap of around 200meV, corresponding to a Mott insulating phase according to DMFT predictions. In addition, by tuning the electron population of the 2D-MOF near-Fermi band structure, via either template induced (work function variation of the pores and wires of the h-BN/Cu(111) electronic moiré) or tip-induced gating (via STM probe sample distance) a Mott metal-insulator transition in the 2D-MOF is proposed.

Even though this work claims the observation of an exotic many-body quantum phenomenon (Mott metal-insulator transition) in a 2D-MOF, the experimental evidence and theoretical support are not strong enough. Therefore, the status of this work is still too preliminary so that it can be published in Nature Communications. The Mott metal-insulator transition should be present in the entire 2D-MOF material (since this effect alters the band structure of the material). Therefore, the present claim that this effect can happen at the local scale (< 5 nm) sounds controversial.

Author reply:

Our dI/dV STS measurements show that: (i) the 2D kagome MOF exhibits an electronic energy gap of ~200 meV (Fig. 2), with occupied and unoccupied band edges following a spatial modulation (from pore to wire regions) given by local variations of the work function resulting from the hBN/Cu(111) moiré pattern (Fig. 3); (ii) when the STM tip is far enough from the surface (such as to not significantly shift the MOF energy levels via the double-barrier tunnelling junction effect), the entire 2D kagome MOF is in a Mott insulating phase, with Mott insulating energy gaps both at the pore and wire regions of the hBN/Cu(111) moiré pattern Fig. 4 and SI Fig. S4); (iii) for smaller tip-sample distances, the wire region exhibits a metallic dI/dV spectrum, with no energy gap and a significant dI/dV signal at the Fermi level, while the pore regions remain Mott insulating (Fig. 3b). We interpret these experimental observations – which are reproduced by our DMFT calculations – as a transition at the wire regions from a Mott insulating phase to a metallic phase; this transition results from the depletion of the kagome MOF electronic bands due to the combination of: (i) large local work function at the wire region (Fig. 3c), and (ii) tip-induced MOF energy level shifts due to the double-barrier tunnelling junction (Fig. 4).

In the Mott insulating phase (i.e., moiré pore regions, and moiré wire regions for large tip-sample distances), electronic states are intrinsically localised at kagome lattice sites due to

strong on-site Coulomb repulsion, with an electronic mean free path that is smaller than the MOF lattice constant. That is, the concept of band structure within conventional band theory (i.e., eigenenergies of single-electron wavefunctions as a function of single-electron wavefunction wavevector k) can be ill defined, with near-Fermi Hubbard 'bands' that can be incoherent, and electronic phenomena are fundamentally local.

Now, at moiré wire regions, when the tip-sample distance is reduced (Fig. 4), we claim that the combined effects of local work function (larger than for the moiré pore regions; Fig. 3c) and double-barrier tunnelling junction lead to a depletion of the localised electronic states of the MOF in the Mott insulating phase (i.e., effectively, a lowering of the MOF chemical potential), within the area of the tunnelling junction (with a typical characteristic length scale of ~ 10 nm given by the tip radius of curvature). This depletion of electronic states leads to the collapse of the Mott insulating phase, into a metallic phase with no gap and an increased density of states at the Fermi level (Fig. 3b, e-g, and Fig. 4d). We acknowledge that this effect occurs within an effective area defined by the tunnelling junction cross section (i.e., typically ~ 10 nm in length), and that electronic confinement at such moiré wire regions can lead to features in the local density of states that are not captured quantitatively by our DMFT calculations.

It is important to note that Mott insulating phases have been observed recently in monolayer 1T-TaS₂ and 1T-TaSe₂, which are 2D crystals with an hexagonal Bravais lattice and a lattice constant of ~ 2 nm, within crystalline domains with characteristic length scales of ~ 10 nm.^{1,2} Importantly, 1T-TaS₂ domains with characteristic length scales of ~ 5 nm can be metallic (i.e., ungapped and with a non-zero density of states at the Fermi level) when strained.³ These phenomena are very similar to the case of our 2D MOF here, and support our interpretation of a Mott insulating phase in monolayer domains with characteristic length scales of ~ 5 -10 nm, with possible local transitions to a metallic phase.

We have now clarified this in our main text discussion (before the conclusion). Our work is not preliminary; it provides a complete characterisation, description and interpretation of our experimental observations, supported by a theoretical formalism (DMFT) which depicts correlated-electron phenomena accurately and reliably (in contrast with other formalisms such as DFT).⁴⁻⁶

Additional experimental evidence, such as temperature dependence studies of the insulating gap size and angle-resolved photoemission spectroscopy measurements are necessary.

Following this suggestion from Reviewer 1, we have now performed dI/dV STS measurements at DCA anthracene extremity sites of the 2D kagome MOF at 77 K, at pore and wire regions of the hBN/Cu(111) moiré pattern (with moiré periodicity similar to the moiré domain in the main text; see below and new Supplementary Fig. S31a). These measurements at 77 K reveal a ~ 200 meV electronic gap at pore regions, and no gap at the Fermi level and an increased Fermi-level dI/dV signal for wire regions (for an intermediate tip-sample distance, as for Fig. 3b in main text), similar to what we observed at 4 K (with some trivial thermal broadening). We also performed further DMFT calculations ($U = 0.65$ eV) of the spectral function, for chemical potentials $E_F = 0.4$ eV (Mott insulating phase) and $E_F = 0.2$ eV (metal-like phase with no gap at the Fermi level), for temperatures between 29 and ~ 600 K (see below and new Supplementary Fig. S32). These DMFT-calculated spectral functions show no significant

changes as the temperature varies within this range (except for trivial thermal broadening), in very good agreement with experiments. Importantly, these supplementary measurements and calculations indicate that: (i) the Mott insulating gap is robust up to high temperatures (in particular room temperature), and (ii) changes in dI/dV spectra are likely to become significant only for temperatures well above room temperature, at which STS measurements are challenging if not possible. We have now added a new Supplementary Section S23 on this matter.

Reviewer 1 also suggests that angle-resolved photoemission spectroscopy (ARPES) measurements might be useful for supporting our claims. We agree that ARPES might be useful as a complementary technique in future studies on systems similar to the one we focus on here. However, ARPES is not adequate for the specific system that we consider here, a single-layer 2D MOF adsorbed on an atomically thin 2D insulator. ARPES does not allow for the detection of unoccupied states (useful for the observation of the Mott insulating energy gap). For our particular sample, k-resolution would be impossible due to the different rotational orientations of the hBN domains on Cu(111), which give rise to different moiré patterns (i.e., there would be multiple crystalline domains within the spot size of an UV light/X-ray source). Moreover, ARPES could lead to radiation-induced damage of the material of interest (in particular of compounds composed of organic molecules as here). In our present case, where the MOF is adsorbed on insulating hBN, irradiation by UV light/X-rays is likely to result in charging (unless an electron flood gun is used), imposing further challenges on ARPES measurements. Importantly, conventional ARPES – a space-averaging technique – cannot provide the real-space resolution necessary for distinguishing MOF electronic properties at pore and wire regions of the hBN/Cu(111) moiré pattern (due to the large UV/X-ray spot size). It would also not allow for measuring MOF energy level shifts via the double-barrier tunnelling junction (DBTJ) effect (for which the STM tip is required), which drive the transitions from Mott insulator to metal at the wire regions. It is well established that Mott energy gaps and Mott insulating phases – including gate-controlled transitions to metallic phases - can be evidenced via dI/dV STS, without the need of ARPES.^{1,3,7-9} Low-temperature dI/dV STS provides very good energy resolution, capability for detecting both occupied and unoccupied states non-invasively, and real-space atomic-scale resolution (useful here for distinguishing MOF electronic properties at wire and pore regions of the hBN/Cu(111) moiré pattern).

Temperature-dependent STS measurements. STS measurements performed at DCA lobe sites of the DCA_3Cu_2 MOF at hBN/Cu(111) moiré pore (orange) and wire (blue) regions at temperatures of (a) 77 K and (b) 4 K. Setpoints: (a) $V_b = -500$ mV, $I_t = 100$ pA; (b) $V_b = -500$ mV, $I_t = 500$ pA (orange), and $V_b = -500$ mV, $I_t = 160$ pA (blue). Black dashed horizontal lines indicate position of $dI/dV = 0$.

Temperature-dependent spectral functions calculated by DMFT. **a**, Spectral function $A(E)$ at different temperatures, with $U = 0.65$ eV, and (a) $E_F = 0.4$ eV (\sim half-filling, corresponding to moiré pore region in experiments) and (b) $E_F = 0.2$ eV (filling corresponding to metal-like phase observed experimentally at wire regions).

In addition, the authors should carefully address the following major concerns that I have (points i to xv).

i) The DCA_3Cu_2 2D-MOF has already been studied on a decoupling layer such as graphene (ref. 31). There a long-range ordered 2D-MOF phase was achieved, by far exceeding the quality of the DCA_3Cu_2 2D-MOF presented in this manuscript. Even though the authors

claim to have achieved a highly crystalline growth, this is not the case so far. The authors must provide better experimental proof to defend this point.

The DCA₃Cu₂ 2D MOF has indeed been studied on graphene before, as acknowledged and referenced explicitly (previous Ref. 31; updated Ref. 33) in our manuscript.¹⁰ This previous study is however fundamentally different from ours: graphene is a semimetal with no energy gap, whose electronic states hybridise with those of the 2D DCA₃Cu₂ MOF (see Fig. 3b of updated Ref. 31), and on which a Mott gap has not been observed. In contrast, it is well established that molecular and metal-organic systems on wide bandgap, single-layer hBN on a metal substrate remain unhybridized with electronic states of the underlying metal substrate, retaining their intrinsic electronic properties (see previous Ref. 30; updated Ref. 31).

We dispute Reviewer 1's claim that the quality of the 2D MOF on graphene in (previous) Ref. 31 (updated Ref. 33) "far exceeded" the quality of the same MOF on hBN/Cu(111) in our current manuscript. Updated Ref. 33 by Yan et al. includes two large-scale STM images of the DCA₃Cu₂ MOF on graphene: a ~40x40nm² image in the main text of a MOF region showing very good crystallinity but which is not defect free (see figure below), and a ~150x150nm² image in the Supplementary Information (SI) with a large amount of presumably unreacted Cu on the surface (bright features in figure below, which the authors fail to explicitly comment on) and with clear domain boundaries. Note that the colour scale and resolution of this latter ~150x150nm² image hinder a reliable evaluation of the overall homogeneity or possible defects of the MOF. Moreover, no Fourier transforms (FTs) of these MOF domain STM images are provided, making a comparison of crystallinity impossible.

Synthesising a large single-crystal MOF is challenging. Here we succeeded in synthesising 2D MOF domains with characteristic lengths of several tens of nm's, with very good quality in terms of monocrystallinity and defect density. This is evidenced by our large-scale STM images (60x60nm² in Fig. 1a, 300x300nm² in SI Fig. S7, and 100x100nm² in SI Fig. S8a), and, importantly, the FTs of these images, showing clear sharp diffraction peaks (inset in Fig. 1a; SI Fig. S8). Note that in our manuscript we explicitly acknowledge the presence of DCA-only domains, and that in the SI we explain that we sacrifice some of the overall MOF yield to obtain high-crystallinity of the MOF domains.

Previous STM and dI/dV STS studies on 2D materials with a lattice geometry similar to that of our MOF here (e.g., single-layer 1T-TaS₂, with a 2D hexagonal lattice and a lattice constant on the order of ~2 nm) have shown a Mott insulating phase on nanoisland domains with characteristic sizes on the order of ~10x10nm².¹ That is, our monocrystalline MOF domains with characteristic lengths on the order of several tens of nm's are sufficiently large to demonstrate a Mott insulating phase.

We have now updated our manuscript, nuancing our previous claim of "highly crystalline growth" and changing it to "monocrystalline growth of the MOF domains". We have now added comments to the main text discussing explicitly the presence of defects in our MOF domains and of DCA-only regions. We have also added a 300x300 nm² image (Fig. S7) to the Supplementary Section S6, highlighting the quality and homogeneity of the 2D MOF, and an SI section on how defects affect the MOF electronic properties (Supplementary Section S11).

We emphasise that the main findings of our manuscript are: (i) the observation of a significant Mott energy gap in a single-layer 2D kagome MOF grown on an atomically thin

insulator, and (ii) the potential for controlling such Mott gap and insulating phase electrostatically. The growth of a perfectly crystalline monolayer 2D kagome MOF on an atomically flat insulator, across areas significantly larger than $\sim 100 \times 100 \text{ nm}^2$, is beyond the scope of our current manuscript and motivates further studies.

Comparison of DCA_3Cu_2 MOF on hBN/Cu(111) (present study) with DCA_3Cu_2 MOF on graphene/Ir(111) (previous Ref. 31; updated Ref. 33). a, b, STM images from (updated) Ref. 33 by Yan et al.¹⁰ c, d, STM images of DCA_3Cu_2 /hBN/Cu(111) in our current manuscript. White arrows: excess Cu; blue arrows: “crack” defects; white dashed circles: vacancy defects.

ii) The authors claim in the abstract that correlated-electron phases, or 2D-MOF “quantum materials” have not been experimentally realized yet. However, in a recent paper, already uploaded into arXiv before this manuscript, Lobo-Checa et al. report 2D magnetism in a very similar 2D-MOF i.e., the DCA_3Fe_2 2D-MOF grown on Au(111) (see <https://arxiv.org/abs/2209.14994>). The authors should give credit to other colleagues working in the same research field. This preprint should be mentioned in the introduction and cited.

We thank Reviewer 1 for drawing our attention to this relevant reference. We have now added it to the introduction as requested (updated Ref. 26). We acknowledge that Lobo-Checa et al. have reported ferromagnetism in the DCA_3Fe_2 MOF, as the result of exchange interactions (i.e., correlations) between unpaired Fe 3d electrons across the organic DCA linkers.¹¹ The main message and physics reported in our manuscript are, however, fundamentally different: the Mott insulating phase results from strong Coulomb interactions between electrons populating the DCA_3Cu_2 MOF near-Fermi kagome bands, which have dominant DCA molecular orbital character, with the Cu 3d shell completely filled (see Kumar et al. DOI: 10.1002/adfm.202106474; updated Ref. 25 in our manuscript)¹². We have now tried to emphasise and clarify this throughout the manuscript.

iii) The synthesis of a 2D-MOF on h-BN via supramolecular coordination chemistry is not new (see ref.30). In addition, the gate control of the DCA₃Cu₂ 2D-MOF electronic structures and charging was also already performed on graphene (ref. 31).

We agree with both Reviewer comments. We do not intend to claim otherwise on either point. We explicitly acknowledge that growth of a 2D MOF (with square lattice) on hBN^{13,14} (previous Ref. 30; updated Refs. 31, 32; updated line 95 of main text), as well as tip-induced gating (i.e., charging via the double-barrier tunnelling junction effect; see Fig. 4a, b in old Ref. 31, now ref. 33; line 247) of unoccupied molecular orbitals of the same DCA₃Cu₂ MOF on graphene, have been achieved previously.¹⁰

We do emphasise, however, that our work represents the first experimental demonstration of: (i) growth of a single-layer 2D MOF with kagome crystal structure on an atomically thin wide bandgap 2D insulator, and (ii) electrostatic control over a Mott insulating phase therein. These findings are fundamentally different from those reported in old Refs. 30, 31.

iv) page 3 (lines 84-85) “a crystalline MOF domain” growth is claimed. However, compared to ref. 31, this is far from crystalline. As shown in Fig. 1(a) of the manuscript, the MOF domain is surrounded by excess DCA molecules, therefore the 3:2 stoichiometry was only successfully achieved on rather small regions of the sample (50 × 50 nm²). In addition, white islands embedded into the 2D-MOF domains may point out to excess Cu islands. Can the authors identify such islands? There are also many additional defects in the MOF domain which are not discussed (black or dark blue points, wires or regions inside the domain). It is therefore almost impossible to find more than two h-BN pore regions together, covered by defect free DCA₃Cu₂ 2D-MOF. The authors should provide experimental evidence of a long-range ordered growth of the 2D MOF. Otherwise, the authors should discuss the defects appearing in the 2D-MOF domains, as well as their limitations to achieve a highly crystalline 2D-MOF growth. This is an important point since such small domain sizes and high defect density will preclude the use of such 2D-MOFs in electronic devices.

As outlined in our response to point i) above, we dispute the Reviewer’s assessment of the comparison of the DCA₃Cu₂ MOF crystallinity between our manuscript here and previous Ref. 31 (updated Ref. 33). Previous Ref. 31 (updated Ref. 33) by Yan et al. provides two large-scale STM images of the DCA₃Cu₂ MOF on graphene: a ~40x40nm² image in the main text of a MOF region showing good crystallinity but which is not defect-free [see Fig. provided in answer to point i)], and a ~150x150nm² image in the SI with a large amount of presumably unreacted Cu on the surface [bright features in panel b of Fig. above in point i), which the authors fail to explicitly comment on] and with clear domain boundaries.¹⁰ No Fourier transforms (FTs) of these MOF domain STM images are provided, making a comparison of crystallinity difficult.

We succeeded in synthesising 2D MOF domains with characteristic lengths of several tens of nm’s, with very good quality in terms of monocrystallinity and defect density. This is evidenced by our large-scale STM images (60x60nm² in Fig. 1a and 100x100nm² in Supplementary Fig S8a), and, importantly, the FTs of these images, showing clear sharp diffraction peaks (inset in Fig. 1a of our manuscript; SI Fig. S8b, c). In our manuscript we explicitly acknowledge the presence of DCA-only domains in these images, and in the SI we explain that we sacrifice some of the overall MOF yield to obtain crystalline MOF domains.

We have now updated our manuscript, nuancing our previous claim of “highly crystalline growth” and changing it to “monocrystalline growth of the MOF domains”. We have now added comments to the main text discussing explicitly the presence of defects in our MOF domains and of DCA-only regions. In the SI, we have also added a Supplementary Section S11 on how such defects might affect the MOF electronic properties, as requested by Reviewer 1.

We have also added a 300x300 nm² image of the DCA₃Cu₂ MOF on hBN/Cu(111) to the Supplementary Section S6 (see Fig. S7). In this image, no DCA-only domains are observed, indicating that the DCA-to-Cu stoichiometry is close to 3-to-2 for this sample preparation (with some Cu clusters, similar to previous Ref. 31 by Yan et al.). The size of this image is ~four times the size of any image in previous Ref. 31, with similar MOF quality despite the insulating substrate (compared to a conductive substrate in previous Ref. 31). The bright islands are indeed excess Cu clusters forming after the final annealing step in our sample preparation; similar bright clusters are also present in previous Ref. 31. We have added a comment to the main text explicitly identifying these as Cu.

We emphasise that the main findings of our manuscript are: (i) the observation of a significant Mott gap in a single-layer 2D kagome MOF grown on an atomically thin insulator, and (ii) the potential for controlling such Mott gap and insulating phase electrostatically. As mentioned above, previous STM and dI/dV STS studies on 2D materials with a lattice geometry similar to that of our MOF here (e.g., single-layer 1T-TaS₂, with a 2D hexagonal lattice and a lattice constant of ~2 nm, similar to our system) have shown a Mott insulating phase on nanoisland domains with characteristic sizes on the order of ~10x10nm² (see updated Ref. 49 in main text by Vano et al.). That is, our monocrystalline MOF domains with characteristic lengths on the order of several tens of nm’s are sufficiently large to support these findings. The growth of a perfectly crystalline monolayer 2D kagome MOFs on an atomically flat insulator, across areas significantly larger than ~100x100nm², is beyond the scope of our current manuscript and requires further studies.

Large-scale STM image of DCA₃Cu₂ MOF on hBN/Cu(111). Image size: 300 × 300nm². $V_b = -1$ V, $I_t = 10$ pA). No DCA-only regions are observed. Sample preparation was close to the desired 3:2 DCA:Cu stoichiometry, with only minor amounts of excess Cu.

v) page 4 (line 87). Here the growth of the 2D-MOF on top of the h-BN/Cu(111) electronic moiré is presented. However, it is known that the moiré periodicity can change approximately from 3 nm to 14 nm on the h-BN/Cu(111) samples grown with borazine in UHV (ref. 30). Is there any moiré periodicity preference for the successful growth of 2D-MOF islands on top? The authors should provide a discussion in this regard.

We thank Reviewer 1 for this valid comment. In our manuscript we focus on the DCA_3Cu_2 MOF grown on hBN/Cu(111) moiré domains with periodicities of ~10-12 nm, since these hBN/Cu(111) regions seem to be the most commonly formed. We do observe examples where the 2D MOF grows on rarer hBN/Cu(111) domains with smaller moiré periodicity (e.g., ~5 nm), where the 2D MOF shows structural and electronic properties similar to those on hBN/Cu(111) moiré domains with a ~10 - 12 nm periodicity, with dI/dV spectra that are consistent with our physical interpretation.

We believe the MOF has no growth preference for particular moiré pattern periodicities. Establishing whether the 2D MOF growth quality depends on the hBN/Cu(111) moiré periodicity is, however, beyond the scope of this manuscript and requires further investigations.

We have now added a comment on this matter in Supplementary Section S6, and a new Supplementary Section S15 including data on the electronic properties of the MOF on hBN/Cu(111) domains with moiré periodicities of ~5 nm and ~10 nm.

vi) page 4 (lines 102-105): The authors should be transparent and mention that DCA_3Cu_2 2D-MOF (the very same systems as the one shown here) has already been grown (as a full monolayer) on top of a rather good decoupling layer of graphene on Ir(111) (ref. 31).

*We have now included an explicit mention of the growth of the DCA_3Cu_2 MOF on graphene/Ir(111) on p. 4, lines 102-103, citing previous Ref. 31 (updated Ref. 33), as requested by Reviewer 1. We do emphasise that the findings of our work are fundamentally different to those of (previous) Ref. 31 by Yan *al.*¹⁰ The electronic states of graphene – a semimetal with no energy gap and with significant electrical conductivity – do show some degree of hybridisation with those of the DCA_3Cu_2 2D MOF (see Fig. 3b of previous Ref. 31). That is, graphene is not a perfectly decoupling layer. In contrast, it is well established that electronic states of molecular and metal-organic systems on wide bandgap, single-layer hBN retain their intrinsic properties and remain unhybridized with substrate states, within a relatively large energy range (~5 eV) around the Fermi level (see previous Ref. 30 by Auwarter *et al.*, updated Ref. 33, and Fig. 1d).¹⁴ We therefore think our growth upon an insulating hBN layer represents a significant advancement towards potential technological applications.*

vii) With respect to Figure 1. The authors only focus on the kagome band structure close to the Fermi level. However, it is known that more kagome bands should appear above and below the energy range considered here (ref. 19). Can the authors comment on this point? Do the authors have DFT calculations for a wider energy range? How does the band structure of the Mott insulating phase look like?

We have added a plot of our DFT calculations for a wider energy range (-3 to 3 eV), for both the free-standing MOF and the MOF on hBN/Cu(111) (see below and new Supplementary Section S1). These calculations, in good agreement with Ref. 19 in the main text, show the three kagome bands near the Fermi level separated from lower-lying occupied and upper-lying empty bands by >1.5 eV. Note that DFT tends to underestimate energy gaps, so this energy separation could be even larger. Because this energy separation is so large, the essential correlated-electron physics can be described by considering solely electronic states near the Fermi level. More DFT calculations can also be found in our previous publication.¹⁵

We have also added to the SI a plot of the energy- and wavevector-dependent spectral function, $A(E, k)$, for the Mott insulating phase, calculated via DMFT ($U = 65$ eV; $E_F = 0.4$ eV, i.e., near half-filling; new Supplementary Section S5). This spectral function shows a significant gap at the Fermi level, with an occupied lower Hubbard band with weakly dispersive features (in particular close to the Γ point), and an empty upper Hubbard band (UHB) with significantly less dispersion. Notably, the diffuse nature of these bands (as opposed to sharp and well-defined, as in conventional band theory) is indicative of band incoherence; this is expected for a Mott insulating phase.

It is important to note that, as mentioned in the main text (lines 125 - 126, p. 5), DMFT captures many-body effects more accurately and reliably than DFT. As such, DMFT is a valid approach for determining the electronic structure of the Mott insulating phase in our case here, and of strongly correlated materials in general.⁴⁻⁶

DFT band structures of MOF over wide energy range. a, Free-standing DCA_3Cu_2 MOF. **b**, DCA_3Cu_2 MOF on hBN/Cu(111). Blue circles: projections onto MOF states.

k -resolved spectral function for free-standing DCA_3Cu_2 MOF, calculated by DMFT ($U = 0.65$ eV, $t = 0.05$ eV, $E_F = 0.4$ eV, corresponding to half-filling).

viii) The authors should provide STS performed on a larger voltage range (for instance from -2V to +2V) to see what happens to the additional kagome bands away from the Fermi level. Is there any strong molecular HOMO/LUMO or 2D-MOF valence/conduction band feature at lower/higher voltages, as already observed in ref. 33 and in A. Kumar et al. Nano Letters 18, 5596 (2018). The latter reference should be added since DCA_3Co_2 2D-MOF was grown on the rather good decoupling graphene layer on Ir(111), whereby a 2D-MOF band structure signature was already observed.

As requested by Reviewer 1, we have now added a new Supplementary Section S12 with a dI/dV spectrum at a MOF DCA lobe site, at a hBN/Cu(111) moiré pore region, on a larger voltage range, from -1 to +1 V (see below). Such a spectrum shows no clear prominent electronic features (e.g., HOMO/LUMO signatures) outside the -0.5 to +0.5 V energy range reported in the main text. This is consistent with our DFT calculations (see above; Supplementary Fig. S1) and those of Ref. 19, which show three kagome bands with dominant DCA LUMO character near the Fermi level, well separated in energy from other fully occupied and completely empty bands lying beyond ~ 1.5 eV below and above the Fermi level, respectively. This is the reason why in our manuscript we focus on the -0.5 to 0.5 V bias voltage range. Note that performing dI/dV STS within a large bias voltage range (e.g., -2 to 2 V) is not trivial due to challenges in preparing a spectroscopically functional STM tip on the MOF/hBN/Cu(111) system.

We have now added A. Kumar et al. Nano Letters 18, 5596 (2018) as a reference (new Ref. 52 in main text), as requested by Reviewer 1.¹⁶ Note that the (non-interacting) band structure of DCA_3Co_2 on graphene is fundamentally different from that of DCA_3Cu_2 on hBN, with the former hosting multiple bands near the Fermi level with very significant contributions from

Co states [e.g., see Fig. 4b of Kumar et al. *Nano Letters* 18, 5596 (2018)].¹⁶ The $DCA_3Cu_2/Cu(111)$ system of (old) Ref. 33 (new Ref. 35) is also fundamentally different from our $DCA_3Cu_2/hBN/Cu(111)$ system, with the MOF interacting significantly with the underlying, more reactive Cu(111) substrate, with significant hybridization between MOF and Cu(111) states, and the near-Fermi kagome bands not half-occupied [see B. Field et al., *npj Computational Materials* 8, 227 (2022)].^{15,17} We therefore assert that a quantitative comparison between the electronic properties of our $DCA_3Cu_2/hBN/Cu(111)$ system, and those of $DCA_3Cu_2/graphene$ and $DCA_3Cu_2Cu(111)$, is not meaningful.

STS measurements over a broader energy window. Orange curve: DCA lobe site of MOF within a pore region of the hBN/Cu(111) moiré pattern. Grey curve: bare hBN/Cu(111) reference spectrum. Spectra normalised and offset for clarity. Orange curve acquired in two parts: setpoint of $V_b = -1$ V, $I_t = 100$ pA for data between -1 V and -500 mV; setpoint of $V_b = -500$ mV, $I_t = 100$ pA for the remaining bias range. Grey curve setpoint: $V_b = -2$ V, $I_t = 100$ pA.

ix) page 6 (line 149) “the high crystalline growth makes a large disorder-related gap unlikely”. If this is the case, how does the STS look at different points of the 2D-MOF island? Are the features identical if one performs STS on the dark pore regions? How do the electronic properties look like at the edge of the 2D-MOF domain? At which position does the domain edge already play a role in disturbing the reported electronic properties?

We thank Reviewer 1 for these valid comments. We have now provided further dI/dV STS measurements at the high-symmetry MOF locations: Cu, DCA lobe, DCA centre and MOF pore sites (to not confuse with what we refer to as a hBN/Cu(111) moiré pore region); see new Supplementary Section S10. The dI/dV spectra at DCA centre and dark MOF pore sites are qualitatively similar to the spectra at adjacent Cu and DCA lobe sites, with spectral features (i.e., lower and upper Hubbard bands) at these Cu and DCA lobe sites stronger. This is consistent with the STM images and dI/dV maps taken at bias voltages corresponding to the lower Hubbard band maximum and upper Hubbard band minimum (see Supplementary Figs. S11-12), showing higher intensity at Cu and DCA lobe sites.

We also added a new Supplementary Section S24 where we compare dI/dV spectra acquired at different pore regions of a hBN/Cu(111) domain with a moiré period of $\lambda \approx 12.5$ nm. The

spectra in Supplementary Fig. S23 are very consistent across six distinct moiré pores. This shows that within the bulk of a MOF domain, the MOF local electronic properties are identical for equivalent sites with respect to the hBN/Cu(111) moiré pattern.

We have also added a new Supplementary Section S11 where we explore the effect of defects (e.g., MOF domain edges and boundaries, vacancies, cracks) on the MOF electronic properties. Supplementary Fig. S14 shows dI/dV STS measurements at the edge of a 2D MOF domain, where DCA molecules are coordinated to only one Cu atom (and not two as within the MOF bulk). At these MOF edge sites, the dI/dV spectra show remnants of the Hubbard bands, with a weaker upper Hubbard band, and an additional peak at ~ 0.6 V, resembling the (uncoordinated) DCA LUMO on hBN [see D. Kumar et al. “Mesoscopic 2D molecular self-assembly on an insulator”, *Nanotechnology* 34, 205601 (2023); updated Ref. 47].¹⁸ Notably, one unit cell away from the MOF domain edge, the MOF local electronic properties are identical to those within the MOF domain bulk.

We have also performed additional dI/dV STS measurements at a defective MOF domain boundary (new Supplementary Fig. S15), qualitatively consistent with dI/dV spectra at the MOF domain edge. Similarly, no significant disturbance in MOF electronic properties is found at defective Cu vacancy sites (new Supplementary Fig. S16).

STS measurements at MOF high-symmetry locations within a pore region of the hBN/Cu(111) moiré pattern. **a**, STM image of DCA₃Cu₂ MOF on hBN/Cu(111) ($V_b = -1$ V, $I_t = 10$ pA). **b**, STS measurements performed at positions corresponding to coloured markers in (a). Grey curve: reference spectrum acquired upon bare hBN/Cu(111). Curves offset for clarity. Dashed lines indicate position of $dI/dV = 0$. Tip height stabilised 150 pm further away from the surface with respect to a setpoint of $V_b = 10$ mV, $I_t = 10$ pA.

dI/dV spectra at different moiré pore regions within hBN/Cu(111) domain with period $\lambda \approx 12.5$ nm. **a**, Cu sites. **b**, DCA lobe sites. All acquisition sites close to centre of moiré pore regions. Spectra normalised and offset for clarity. Pore 1 setpoint: $V_b = -500$ mV, $I_t = 500$ pA. Pores 2,3 setpoints: 190 pm further from the surface with respect to setpoint of $V_b = 10$ mV, $I_t = 10$ pA. Pores 4,5 setpoint: 225 pm further from the surface with respect to setpoint of $V_b = 10$ mV, $I_t = 10$ pA. Pore 6 setpoint: 150 pm further from the surface with respect to setpoint of $V_b = 10$ mV, $I_t = 10$ pA.

STS measurements at the edge of a DCA_3Cu_2 MOF domain. **a**, STM image of the edge of a MOF domain ($V_b = -1$ V, $I_t = 10$ pA). **b**, STS measurements at Cu sites (circle markers) and DCA lobe sites (triangle markers) both at the edge of the MOF domain (black) and within the MOF domain (red). Curves offset for clarity. Setpoints: $V_b = -500$ mV, $I_t = 100$ pA.

STS measurements at a DCA_3Cu_2 MOF domain boundary. **a**, STM image showing a MOF domain boundary ($V_b = -1$ V, $I_t = 10$ pA). **b**, STS measurements at DCA lobe sites both at the MOF domain boundary (black) and within the MOF domain (red). Curves offset for clarity. Setpoints: $V_b = -500$ mV, $I_t = 100$ pA.

STS measurements at Cu vacancy defect within a DCA_3Cu_2 MOF domain. **a**, STM image showing a Cu vacancy defect ($V_b = -1$ V, $I_t = 10$ pA). **b**, STS measurements at Cu sites in proximity of Cu vacancy (green and orange), and at Cu vacancy (blue). Curves offset for clarity. Setpoints: 135 pm further away from the surface with respect to a setpoint of $V_b = 10$ mV, $I_t = 10$ pA.

x) With respect to Figure 2. Why are the spectra asymmetric? Why not show dI/dV maps at particular energies instead of STM images?

We thank Reviewer 1 for this valid question. Indeed, the dI/dV signal at the DCA anthracene extremity site, near the centre of a hBN/Cu(111) moiré pore region, for $V_b = \sim -0.2$ V (corresponding to the lower Hubbard band) is smaller than the dI/dV signal at the same location for $V_b = \sim 0.2$ V (corresponding to the upper Hubbard band). Conversely, the dI/dV signal at the Cu site for $V_b = \sim -0.2$ V is larger than the dI/dV signal at the same location for $V_b = \sim 0.2$ V (Fig. 2a in main text). We have now added to Supplementary Fig. S9 the sum of these two local dI/dV spectra at DCA lobe and Cu sites, showing lower and upper Hubbard band peaks that are almost perfectly symmetric, consistent with the DMFT-calculated spectral function. It is important to note that the DMFT calculations do not capture local variations of the spectral function (Fig. 1e, Fig. 3d in main text; updated Supplementary Fig. S9); it is more meaningful to compare the DMFT-calculated spectral function to the spatial integration of locally acquired dI/dV spectra.

We speculate that the local variations in dI/dV spectra might be related to the spatial distribution and orbital character of the states associated with the lower and upper Hubbard bands, with Cu atoms and DCA cyano groups contributing plausibly more to the lower Hubbard band, and DCA anthracene extremities contributing more to the upper Hubbard band. Note that nontrivial spatial distribution and orbital texture of electronic states in 2D Mott insulators have already been observed, with such effects poorly captured by theory.¹⁹ We have now added a comment on this matter to Supplementary Section S7.

We show dI/dV maps in Supplementary Fig. S12. These maps represent a single cut in energy. With states of interest shifting in energy as a function of position with respect to the hBN/Cu(111) moiré pattern (Fig. 3 in main text), these maps show characteristic electronic features within small regions. Therefore, in the main text we show STM topographic images (Fig. 2b-e), which have contributions from electronic states with eigenenergies between the specified bias voltage and the Fermi level. As a result, we claim that, on the hBN decoupling layer, these topographic STM images resolve the spatial distribution of the electronic states of interest, faithfully and more clearly. We have updated Supplementary Fig. S12 by adding supplementary dI/dV maps at positive bias voltages. A discussion and justification of our choices can also be found in Supplementary Section S9.

Comparison between experimental dI/dV spectra and DMFT spectral function for DCA_3Cu_2 MOF at a hBN/Cu(111) moiré pore region. **a**, Experimental dI/dV spectra, as shown in Fig. 2a of main text. Upper (lower) curve acquired at DCA lobe (Cu) site (tip-sample distance defined by setpoint $V_b = -500$ mV, $I_t = 500$ pA). **b**, Sum of the two spectra in (a). **c**, Spectral function, $A(E)$, predicted by DMFT, as shown in Fig. 1e of main text [$U = 0.65$ eV; E_F chosen to match electron filling predicted by DFT for MOF on hBN/Cu(111)]. Asymmetry of spectra in (a) with respect to energy may be indicative of differences in spatial distribution of electronic states associated with LHB and UHB.

xi) page 6 (lines 157-162): Figure 3 should be explained in more detail in the manuscript, specially panels 3e and 3f. The order of the figure description should be changed in the text (a, b, c...). Regarding panel 3b, apart from the theoretically predicted peak at the Fermi level, which can not be captured experimentally, there are many additional peaks in the unoccupied region (>0.2 V). These peaks or modulations are not present in the theoretical simulations. What are they? They have a similar amplitude to the relevant peaks close to the Fermi level, therefore, most probably are not artefacts. The authors should describe them.

We thank Reviewer 1 for this comment. On p. 6 (l. 157-162), we referred to Fig. 3e, f when we actually meant Fig. 3b, d. This was a typographical error from our part. We have now corrected this in the main text and explained Fig. 3 more in detail.

Indeed, the dI/dV spectra in Fig. 3b show supplementary peaks for $V_b > \sim 0.2$ V, which are not predicted by the DMFT calculations (Fig. 3d). We thank Reviewer 1 for this observation.

These experimental dI/dV peaks for $V_b > \sim 0.2$ V are more prominent at locations close to the wire regions, where dI/dV features due to tip-induced charging via the double-barrier tunnelling junction (DBTJ) effect were observed (see Fig. 4). We therefore propose that these supplementary dI/dV peaks are the result of tip-induced charging via the DBTJ effect, as observed for moiré wire regions and described in Fig. 4. The DMFT calculations do not consider the DBTJ or tip-induced effects; therefore, they cannot predict such features. We have now added a comment on this matter in the main text, and a new Supplementary Section S17.

We re-emphasise that the main messages of our manuscript are: (i) the observation of a significant Mott gap in a single-layer 2D MOF, and (ii) the electrostatic control of such Mott gap and insulating phase. As such, we focus on dI/dV features at energies close to the Fermi level.

Charging features at a pore-wire boundary of the moiré pattern. **a**, STM image of MOF/hBN/Cu(111) showing two pore regions and one wire region of the hBN/Cu(111) moiré pattern ($V_b = -1$ V, $I_t = 10$ pA). **b**, dI/dV spectra for different tip-sample changes (Δz), acquired at MOF Cu site, as indicated in (a). Spectra normalised and offset for clarity. Setpoints range between 250 pm further from surface (bottom curve) to 105 pm further from surface (top curve) with respect to a setpoint of $V_b = 10$ mV, $I_t = 10$ pA.

xii) Figure 3a,b. The authors perform STS at Cu adatom positions. For completeness, the authors should also perform the same STS sequence at the DCA anthracene lobe positions. Does the same Mott metal-insulator transition appear or evolve in the same fashion as on the Cu adatoms?

We thank Reviewer 1 for this suggestion. We have now added further dI/dV spectra acquired at DCA lobe sites, for different locations within the MOF/hBN/Cu(111) domain considered in

Fig. 3 of the main text, from moiré pore to wire to pore regions, similar to Fig. 3a, b (see new Supplementary Section S14). We observe the exact same trend as for the Cu sites, that is, a ~ 200 meV gap at the Fermi level and nearly null Fermi-level dI/dV signal at pore regions, and no gap at the Fermi level and a significant increase of Fermi-level dI/dV signal at the wire region (new Supplementary Fig. S19). We believe this revision strengthens our arguments.

STS measurements at DCA lobe sites of DCA₃Cu₂ MOF at different locations of the hBN/Cu(111) moiré pattern. **a**, STM image of MOF ($V_b = -1$ V, $I_t = 10$ pA). White dashed circles (P): hBN/Cu(111) moiré pore regions, separated by wire region (W). Grey arrow indicates moiré period $\lambda \approx 12.5$ nm. **b**, dI/dV spectra acquired at MOF DCA lobe sites, at positions indicated by coloured markers in (a). Tip 190 pm further from surface with respect to setpoint $V_b = 10$ mV, $I_t = 10$ pA. Energy gap $E_g \approx 200$ meV Fermi level at P regions, with no Fermi-level gap at W region (LHBM: lower Hubbard band maximum; UHBM: upper Hubbard band minimum). **c**, Selection of spectra in (b), plotted for clarity. **d**, dI/dV signal at Fermi level ($V_b = 0$) as a function of position x with respect to hBN/Cu(111) moiré pattern, for each spectrum in (b). **e**, **f**, LHBM (squares), UHBM (circles) and energy gap E_g (triangles), as a function of x position, extracted from (b).

xiii) Figure 3. Since the Mott metal-insulator transition is highly dependent on the work function variation of the h-BN/Cu(111) electronic moiré, the authors should provide more experimental evidence that this effect can be reproduced for different moiré periodicities.

We thank Reviewer 1 again for another great suggestion. We have performed further measurements for the DCA₃Cu₂ MOF on hBN/Cu(111) domains with different moiré periodicities of ~ 5 nm ~ 10 nm (in addition to the data in the main text for a moiré periodicity of ~ 12.5 nm). See images below and new Supplementary Section S15. For these smaller moiré periodicities, the dI/dV spectra acquired from pore to wire regions all exhibit a trend qualitatively similar to that for the 12.5 nm moiré periodicity in the main text. For these smaller moiré periodicities, the energy shift of the Hubbard bands as a function of position

along the moiré pattern is reduced, consistent with the expected variation of the local work function of hBN/Cu(111) with respect to hBN/Cu(111) moiré periodicity²⁰ (see new Supplementary Fig. S22). This further corroborates and strengthens the interpretation of our observations.

STS measurements at Cu sites of MOF on hBN/Cu(111) domain with moiré period $\lambda \approx 10$ nm. **a**, STM image of MOF ($V_b = -1$ V, $I_t = 10$ pA). White dashed circles (P): hBN/Cu(111) moiré pore regions, separated by wire region (W). Grey arrow indicates moiré period. **b**, dI/dV spectra acquired at MOF Cu sites, at positions indicated by coloured markers in (a). Tip 190 pm further from surface with respect to setpoint $V_b = 10$ mV, $I_t = 10$ pA. Energy gap $E_g \approx 200$ meV at Fermi level for P regions, vanishing at W region (LHBM: lower Hubbard band maximum; UHBM: upper Hubbard band minimum). **c**, dI/dV signal at Fermi level ($V_b = 0$) as a function of position x on hBN/Cu(111) moiré pattern, extracted from spectra in (b), showing increased signal within W region. **d**, **e**, LHBM (squares), UHBM (circles) and energy gap E_g (triangles) as a function of x position, extracted from (b). Black line in (d): expected LHBM variation due to local moiré work function modulation for this moiré period.²⁰ Grey shaded area: uncertainty.

STS measurements at Cu sites of MOF on hBN/Cu(111) domain with moiré period $\lambda \approx 5$ nm. **a**, STM image of MOF ($V_b = -1$ V, $I_t = 10$ pA). White dashed circles (P): hBN/Cu(111) moiré pore regions, separated by wire region (W). Grey arrow indicates moiré period. **b**, dI/dV spectra acquired at MOF Cu sites, at positions indicated by coloured markers in (a). Tip 190 pm further away from surface with respect to setpoint $V_b = 10$ mV, $I_t = 10$ pA. Energy gap $E_g \approx 200$ meV at Fermi level for P regions, vanishing at W region (LHBM: lower Hubbard band maximum; UHBM: upper Hubbard band minimum). **c**, dI/dV signal at the Fermi level ($V_b = 0$) as a function of position x on hBN/Cu(111) moiré pattern, extracted from spectra in (b), showing increased signal within W region. **d**, **e**, LHBM (squares), UHBM (circles) and energy gap E_g (triangles) as a function of x position, extracted from (b). Black line in (d): expected LHBM variation due to local moiré work function modulation for this moiré period.²⁰ Grey shaded area: uncertainty.

Comparison between local work function modulation and LHBM variation for different moiré periods. Black markers: local work function shift (and uncertainties) for different moiré periods, from

prior literature.²⁰ Red markers: Experimentally observed variations in LHBM energy for the three different moiré periods considered in this work.

xiv) In Figure 4 the STS were performed at a DCA anthracene lobe position. This point is not mentioned in the manuscript and is important. For completeness, the authors should perform the gating experiment on a Cu adatom position.

We thank Reviewer 1 for this suggestion. We have specified for Fig. 4 that the dI/dV spectra were acquired at DCA anthracene lobe sites. We have also added dI/dV spectra at Cu sites, for both pore and wire moiré regions, for different tip-sample distances Δz . These spectra are consistent with those acquired at DCA lobe sites. See new Supplementary Section S21.

Importantly, the DBTJ model fitting parameters are highly consistent between DCA lobe measurements and Cu site measurements, for both moiré pore and wire regions. See new Supplementary Section S21 and Supplementary Table 1. This further validates our overall interpretation of the experimental observations.

Tip-induced gating at a Cu site of the MOF within a wire region of the hBN/Cu(111) moiré pattern. **a**, dI/dV spectra at MOF Cu site, for different $\Delta z + z_0$ (z_0 given by STM setpoint $V_b = 10$ mV, $I_t = 10$ pA). Purple circles (red squares): MOF charging peak (intrinsic electronic state at MOF band edge, respectively). Spectra normalised and offset for clarity. **b**, V_{charge} [purple circles in (a)] and V_{state} [red squares in (a)] as a function of Δz . Black solid lines: global fits to Eqs. (1) and (2) in main text. **c**, dI/dV signal at Fermi level ($V_b = 0$) as a function of Δz , from (a). Increased dI/dV ($V_b = 0$) indicates metallic phase. Inset: STM image with position where $dI/dV(\Delta z)$ were performed indicated by blue circle ($V_b = -1$ V, $I_t = 10$ pA).

dI/dV spectroscopy for Cu site of MOF within a pore region of hBN/Cu(111) moiré pattern. **a**, dI/dV spectra for different tip-sample distances $\Delta z + z_0$ (z_0 represents tip-sample distance for STM setpoint of $V_b = 10$ mV, $I_t = 10$ pA) at a Cu site [location indicated by red circle in (c)]. Red squares (circles) indicate LHBM (UHBM, respectively). **b**, LHBM and UHBM as a function of Δz , from (a). Black dashed curves: fits based on Eqs. (S7) and (S8). **c**, STM image of DCA₃Cu₂ MOF on hBN/Cu(111). Red circle indicates position where spectra shown in (a) were acquired ($V_b = -1$ V, $I_t = 10$ pA).

xv) Figure S10. It is not clear at all that the charging rings are increasing their perimeter around the DCA molecule (see ref. 31). It looks just the same intensity at the Cu positions. This is not consistent with the molecular charging ring features reported in ref. 31. These dI/dV maps do not support the interpretation of charging peaks given in Figure 4 of the manuscript.

We thank Reviewer 1 for this comment and we agree that our previous Fig. S10 was not convincing. We have now updated this figure (note that it is now Supplementary Fig. S23), showing new dI/dV maps at positive bias voltages which more clearly show a charging ring, evolving from the centre of a DCA molecule within the MOF at a wire region of the hBN/Cu(111) moiré pattern, and expanding radially with increasing bias. Previous Ref. 31 by Yan et al. (updated Ref. 33) includes dI/dV maps of the 2D DCA₃Cu₂ MOF on graphene, showing charging rings around some (and not all) DCA molecules for negative bias voltages (Fig. 4e-j in previous Ref. 31).¹⁰ These charging rings shrink with increasing bias voltage (i.e., with decreasing absolute value of bias voltage). At positive biases, we expect the opposite trend: charging rings should increase with increasing bias voltage. This is consistent with our new Supplementary Fig. S23.

dI/dV maps of $\text{DCA}_3\text{Cu}_2/\text{hBN}/\text{Cu}(111)$: charging ring. **a**, STM image of DCA_3Cu_2 at a wire region of $\text{hBN}/\text{Cu}(111)$ moiré pattern ($V_b = -1$ V, $I_t = 10$ pA). **b-m**, dI/dV maps of region in (a), at indicated bias voltages V_b , obtained via numerical derivative of pixel-by-pixel $I(V)$ curves. At each pixel, the tip-sample distance was stabilised 300 pm further away from the surface relative to a setpoint of $V_b = 10$ mV, $I_t = 10$ pA, before $I(V)$ acquisition. Scale bars: 1 nm.

Reviewer 2

In their manuscript entitled “Gate control of Mott metal-insulator transition in a 2D metal-organic framework”, B. Lowe et al claim that in a metal-organic network grown on an $\text{hBN}/\text{Cu}(111)$ substrate it is possible to control a metal-insulator transition through two mechanisms. One of them is based in the modulation of the surface potential due to the presence of a moiré pattern between hBN and $\text{Cu}(111)$. In the second part of the manuscript, they control the transition by varying the sample tip distance. There are different aspects of the article that need to be clarified before the article can be published.

Figure 3 shows the spectra measured by moving the tip between the areas of the moiré called pores to the areas called wires. The experimental spectra show a modulation in the position of the bands depending on the moiré area. Upon reaching the areas called wires, the gap disappears and very intense peaks extend from the Fermi level up to +0.4eV. These spectra features are not mentioned or discussed in the manuscript.

We thank Reviewer 2 for this valid comment. Indeed, the experimental dI/dV spectra in Fig. 3b for locations near the $\text{hBN}/\text{Cu}(111)$ moiré wire region show significant peaks between the Fermi level and $V_b \sim 0.4$ V. At such locations close to the wire regions, we observed dI/dV

features due to tip-induced charging via the double-barrier tunnelling junction (DBTJ) effect; see Fig. 4. We therefore propose that these dI/dV peaks in Fig. 3b could be the result of tip-induced charging via the DBTJ effect, as observed for moiré wire regions and described in Fig. 4. Note that the DMFT calculations do not consider the DBTJ or tip-induced effects; therefore, they cannot predict such features. We have now added a paragraph on this matter in the main text, and a related new Supplementary Section S17.

We want to emphasise that the main messages of our manuscript are: (i) the observation of a significant Mott gap in a single-layer 2D MOF, and (ii) the electrostatic control of such Mott gap and insulating phase. As such, in the main text we focus on dI/dV features at energies close to the Fermi level.

Charging features at a pore-wire boundary of the moiré pattern. **a**, STM image of MOF/hBN/Cu(111) showing two pore regions and one wire region of the hBN/Cu(111) moiré pattern ($V_b = -1$ V, $I_t = 10$ pA). **b**, dI/dV spectra for different tip-sample changes (Δz), acquired at MOF Cu site, as indicated in (a). Spectra normalised and offset for clarity. Setpoints range between 250 pm further from surface (bottom curve) to 105 pm further from surface (top curve) with respect to a setpoint of $V_b = 10$ mV, $I_t = 10$ pA.

The calculations reproduce the energy position of the bands observed in the experiment until the wire areas where the disagreement with the experiments it is clear. The calculations corresponding to the wire areas show very pronounced and very narrow peaks. The authors attributed their origin to coherent quasiparticle. Contrary to what the authors say in the manuscript, the calculations do not show a gap collapse in the wire areas, on the contrary it becomes wider and extends to almost $+0.4$ eV.

According to the authors, the control of the metal-semiconductor transition induced by the substrate is deduced from the comparison between the experimental results and the

theoretical calculations shown in Figure 3. Before this can be stated it is necessary to discuss in detail the discrepancies mentioned above between theory and experiments.

We thank Reviewer 2 for these very valid comments, with which we agree. Indeed, in the main text (previously, p. 7, line 181), we claimed that the DMFT calculations show, for a chemical potential corresponding to the moiré wire region, a collapse of the energy gap E_g . What we meant is that, for such a chemical potential (i.e., smaller than that for a moiré pore region), the DMFT calculations show no gap at the Fermi level, with a non-zero increased spectral function (indicating the presence of electronic states) at the Fermi level. This is consistent with our experiments, which show a larger Fermi-level dI/dV signal at the moiré wire region in comparison to the moiré pore region (for the considered tunnelling conditions; see updated Fig. 3e and new Supplementary Sections S14-15), and with our claim of a metallic phase at the moiré wire region (i.e., for tunnelling parameters, in particular tip-sample distance, used in Fig. 3). This increase in Fermi-level DMFT-calculated spectral function and measured dI/dV marks the onset of the Mott energy gap collapse, and of the transition from Mott insulating phase to metal (see Supplementary Fig. S2b).

As pointed out by Reviewer 2, the DMFT calculations for a chemical potential corresponding to the moiré wire region show a pronounced narrow peak near the Fermi level. These peaks are not observed in our experimental dI/dV curves (Fig. 3). We explain this discrepancy between theory and experiment by the fact that: (i) the DMFT calculations assume a uniform chemical potential for an infinite system, omitting effects of locality; this assumption is reasonable for the Mott insulating phase (i.e., localised states), but can lead to unaccounted spectroscopic features for the metallic phase in the finite wire due to long electronic coherence lengths and electronic confinement (as mentioned in the main text methods section related to DMFT calculations, and in the updated Supplementary Sections S2, S3); (ii) limitations of the analytic continuation approach used in the DMFT calculations result in spurious features (dips, peaks) in the DMFT-calculated spectral function at the Fermi level for the metallic phase (as mentioned in the Supplementary Fig. S2 caption).

Despite these important assumption and limitations, we emphasise that experimental and theory agree very well with the observation of the ~ 200 meV Mott gap, the energy modulation of the lower and upper Hubbard bands, and the increase in Fermi-level density of states, with no gap at the Fermi level (i.e., metallic phase, indicating the onset of the Mott gap collapse as the population of the kagome bands decreases), for a variation of the chemical potential in the DMFT calculations that follows quantitatively the variation of the local work function given by the hBN/Cu(111) pattern in the experiments.

We have now updated the main text, clarifying what we mean by ‘gap collapse’, and providing a more detailed discussion on the discrepancies between experiment and theory.

In the second part, the authors show how the metal-insulator transition can be induced by changing the tip-sample distance. To do this, they carry out experiments in one of the areas called wire. Figure 4d shows how in this area the measured spectra may or may not show a gap depending on the tip-sample distance. This result somehow invalidates the manuscript's first claim that moiré registration controls the existence of a gap or not. It seems that the gap collapse depends on the parameters used to perform the measurements.

We thank Reviewer 2 for this valid comment. We acknowledge that our main text was perhaps not perfectly clear on this point. Both the considered MOF region relative to the underlying hBN/Cu(111) moiré pattern and the tunnelling parameters (i.e., tip-sample distance, bias voltage) determine the energy level alignment and population of the MOF electronic states, and hence the corresponding electronic phase (i.e., Mott insulator or metal) of the MOF region at the STM junction.

It is important to note that: (i) the MOF at the hBN/Cu(111) moiré pore regions exhibits a Mott gap and is in the Mott insulating phase regardless of the tunnelling parameters, (ii) the MOF at the hBN/Cu(111) moiré wire regions exhibits a Mott gap and is in the Mott insulating phase for large tip-sample distances (i.e., when MOF energy level shifts due to the double-barrier tunnelling junction are negligible), and (iii) the MOF at the hBN/Cu(111) moiré wire regions is in a metallic phase (with no energy gap at the Fermi level, and with a significant non-zero Fermi-level density of states) for intermediate tip-sample distances (i.e., when MOF energy level shifts due to the double-barrier tunnelling junction result in partial depopulation of MOF electronic states).

We have revised our main text to make this clear.

After reading the manuscript and the supplementary material, it is not clear to me how the assignments of the purple circles and red squares are made in Figure 4 or how it is decided whether the observed gap corresponds to a trivial insulator or a Mott insulator.

We thank Reviewer 2 for this valid comment. We agree that the explanation for the assignments of the purple and red markers, and of the observed gaps to a trivial or Mott insulator in Fig. 4, could be clearer in our manuscript.

The dI/dV spectra for the MOF at the hBN/Cu(111) moiré wire region in Fig. 4d show an electronic gap, with a clear peak (sharper than the near-Fermi band features in Fig. 3b, with a maximum indicated by the purple circles) at positive bias voltage for large tip-sample distances, and at negative bias voltage for small tip-sample distances. These spectra also show a subtler band edge (indicated by the red squares, similar to the near-Fermi band features in Fig. 3b; see updated Supplementary Section S8 for information on the determination of band edges) at a bias voltage of sign opposite to that of the sharp peak, i.e., at negative bias voltage for large tip-sample distances and at positive bias voltage for small tip-sample distances. For intermediate tip-sample distances, these spectra are gapless, with a significantly larger Fermi-level dI/dV signal (see Fig. 4f). From these observations, we infer that the MOF is an insulator (trivial or Mott) for large and small tip-sample distances (i.e., dI/dV spectra with a gap at the Fermi level), and a metal for intermediate tip-sample distances (i.e., dI/dV spectra with no gap at the Fermi level).

The bias voltage position of the sharp peak (purple circles) increases linearly with respect to tip-sample distance, whereas the bias voltage position of the subtler band edge decreases nonlinearly with tip-sample distance. Notably, Eqs. 1, 2 of the main text (now on p. 10), corresponding to the double-barrier tunnelling junction (DBTJ) model, provide very good fits for the bias voltage positions of both sharp peak and subtle band edge as a function of tip-sample distance (black curves in Fig. 4e). This provides compelling evidence that the sharp peaks (purple circles) are associated with charging of an intrinsic MOF electronic state lying at the edge (red squares) of a fully populated (large tip-sample distances) or completely

empty (small tip-sample distances) band (see cartoon schematics in Fig. 4a-c and now Supplementary Fig. S26, and previous Refs. 33, 43, 45, now Refs. 33, 46, 48).

Now, dI/dV spectra for the MOF at a pore region of the hBN/Cu(111) moiré pattern show a ~ 200 meV electronic energy gap at the Fermi level (Fig. 2). These spectra resemble the spectral function of the 2D kagome MOF in the Mott insulating phase (Fig. 1e), calculated via DMFT with a chemical potential that is consistent with the DFT-predicted occupation of the near-Fermi kagome bands for the MOF on hBN/Cu(111) (Fig. 1d, Supplementary Fig. S1). At an adjacent moiré wire region, the local work function increases by ~ 0.2 eV for the specific periodicity of the MOF/hBN/Cu(111) domain considered (Fig. 3c; see new Ref. 29-31). Accordingly, the near-Fermi electronic states of the MOF at this moiré wire region are shifted upwards in energy in comparison to the near-Fermi electronic states of the MOF at the moiré pore region. The dI/dV spectra in Fig. 3b for this moiré wire region (for the specific tunnelling parameters used) show no gap at the Fermi level, with a significant non-zero Fermi-level dI/dV signal (new Fig. 3e; Supplementary Fig. S19), indicative of a metallic phase. These experimental dI/dV spectra are consistent with the spectral function of the MOF calculated via DMFT for a chemical potential that is reduced (in comparison with the DMFT calculations for the moiré pore region); such DMFT spectral function exhibits a significant magnitude and no gap at the Fermi level, indicating a metallic phase, resulting from the depopulation of the MOF bands (Fig. 3d and Supplementary Fig. S2).

From these observations we infer that: (i) the MOF at the moiré pore region is in a Mott insulating phase, and (ii) the MOF at the adjacent moiré wire region, for the specific tunnelling parameters used (i.e., bias voltage, tip-sample distance) in Fig. 3b, is in a metallic phase (as the result of the depopulation of the MOF near-Fermi electronic states due to the increase in local work function).

Now, let us consider the MOF at such a moiré wire region, which is in the metallic phase (i.e., for specific tunnelling parameters). The work function of the STM tip is larger than that of the sample (Supplementary Section S19). When the tip-sample distance is reduced, the double-barrier tunnelling junction (DBTJ) leads to an upward energy shift of the MOF electronic states [with respect to the Cu(111) Fermi level]: the MOF electronic states become further depopulated (Supplementary Fig. S26). That is, when the dI/dV spectra in Fig. 4d transition from gapless (metallic) to gapped (insulator) as the tip-sample distance decreases, we infer that the MOF electronic states associated with the MOF kagome bands become empty, with the Fermi level lying below the bottom of the three kagome bands (see DFT band structure over a wide energy range in new Supplementary Fig. S1). From this, we associate the gapped dI/dV spectra at the top of Fig. 4d (for small tip-sample distances) to a trivial insulating phase of the MOF.

Let us again consider the MOF in the metallic phase at such a moiré wire region (i.e., gapless spectra in Fig. 4d at intermediate tip-sample distances). When the tip-sample distance is now increased, the DBTJ leads to a downward energy shift of MOF electronic states [with respect to the Cu(111) Fermi level], which become further populated (Supplementary Fig. S26). That is, when the dI/dV spectra in Fig. 4d transition from gapless (metallic) to gapped (insulator) as the tip-sample distance increases, we infer that the population of the MOF electronic states associated with the MOF kagome bands also increases in turn, reaching a threshold that opens the Mott gap, with the Fermi lying in such Mott gap. This population of MOF states (here driven by the DBTJ effect and the increase in tip-sample distance) is analogous to the population of MOF states at the adjacent moiré pore

region (due to the smaller local work function at such a moiré pore region; Fig. 3c). From this, we associate the gapped dI/dV spectra at the bottom of Fig. 4d (for large tip-sample distances) to the Mott insulating phase of the MOF.

This inference of a Mott gap at the moiré wire region for large tip-sample distances is supported by DMFT. Indeed, when the chemical potentials used in the DMFT calculations of Fig. 3d are all offset upwards by 45 meV (i.e., leading to further population of the MOF electronic states, mimicking the effect of a tip-sample distance increase), the metallic spectral functions associated with the moiré wire region (with no gap at the Fermi level) all become Mott gapped (see Supplementary Fig. S4).

We have now revised the main text and Supplementary Section S19, including details on how the purple circles and red squares in Fig. 4 were assigned, and on how we associated the observed energy gaps to either a trivial or Mott insulator.

No details are given as to whether the energy levels shown in Figure S12 are a cartoon or the result of a calculation. This detail is important to understand how the authors identify the gap character.

We thank Reviewer 2 for this comment. The schematics in Fig. 4a-c and in previous Supplementary Fig. S12 (now Supplementary Fig. S26) are qualitative, cartoon illustrations of the MOF energy level shifts that result from tip-sample distance variations and the double-barrier tunnelling junction (DBTJ) effect.

We have now clarified this in the captions of Fig. 4a-c, methods section, and Supplementary Fig. S12 (now Supplementary Fig. S26).

Reviewer 3

In this work, the authors reported the experimental synthesis and characterization of a single-layer 2D DCA₃Cu₂ MOF on a wide bandgap BN substrate, which is further shown to host a robust Mott insulating phase and can achieve a Mott metal-insulator transition using electrostatic control. The experimental observations are quantitatively consistent with theoretical predictions (both DMFT calculations and DBTJ model). Direct experimental measurements on a Mott metal-insulator transition in 2D MOFs remain elusive. One of the challenges lies in the experimental synthesis of large single-crystal MOF samples. It is nice to see that the authors have succeeded in making one such sample and successfully demonstrated the Mott metal-insulator transitions induced via either template or tip.

We thank Reviewer 3 for their comment and careful reading of our manuscript.

However, it is important for the authors to address a question that arises from the examination of the large-scale samples, as depicted in Figures 1a and S4. One can clearly see structural defects, such as vacancies and grain boundaries. As these defects may have a profound influence on the electronic properties of the MOF, it would be good for the authors to have some discussion about the potential impact of these bulk defects. This work represents a noteworthy contribution to the research field of 2D MOFs, shedding light on elucidating the

Mott metal-insulator transition in MOFs. I would recommend its publication in Nature Communications after the authors address the aforementioned concerns.

We thank Reviewer 3 for this valid suggestion which has helped improve our manuscript. We have added a comment in the main text and a new Supplementary Section S11 focussed on the electronic properties of the MOF at defect sites (e.g. vacancies, domain grain boundaries) and at boundaries of 2D MOF domains. At these sites, dI/dV spectra show remnants of the Hubbard bands, with a weaker upper Hubbard band and an additional peak at ~ 0.6 V, resembling the LUMO of the (uncoordinated) DCA molecule on hBN [see D. Kumar et al. "Mesoscopic 2D molecular self-assembly on an insulator", Nanotechnology 34, 205601 (2023); Ref. 47 in updated main text].¹⁸ The influence of these defects on the MOF electronic properties is very local, however: a short distance away (\sim one MOF unit cell) from one of such defect sites or domain boundaries, the local MOF electronic properties are identical to those within a defect-free MOF bulk region (new Supplementary Figs. S14-16). We therefore claim that the presence of these defects does not alter the main message of our manuscript.

STS measurements at the edge of a DCA_3Cu_2 MOF domain. **a**, STM image of the edge of a MOF domain ($V_b = -1$ V, $I_t = 10$ pA). **b**, STS measurements at Cu sites (circle markers) and DCA lobe sites (triangle markers) both at the edge of the MOF domain (black) and within the MOF domain (red). Curves offset for clarity. Setpoints: $V_b = -500$ mV, $I_t = 100$ pA.

STS measurements at a DCA_3Cu_2 MOF domain boundary. **a**, STM image showing a MOF domain boundary ($V_b = -1$ V, $I_t = 10$ pA). **b**, STS measurements at DCA lobe sites both at the MOF domain boundary (black) and within the MOF domain (red). Curves offset for clarity. Setpoints: $V_b = -500$ mV, $I_t = 100$ pA.

STS measurements at Cu vacancy defect within a DCA_3Cu_2 MOF domain. **a**, STM image showing a Cu vacancy defect ($V_b = -1$ V, $I_t = 10$ pA). **b**, STS measurements at Cu sites in proximity of Cu vacancy (green and orange), and at Cu vacancy (blue). Curves offset for clarity. Setpoints: 135 pm further away from the surface with respect to a setpoint of $V_b = 10$ mV, $I_t = 10$ pA.

References:

1. Vaňo, V. *et al.* Artificial heavy fermions in a van der Waals heterostructure. *Nature* **599**, 582–586 (2021).
2. Fei, Y., Wu, Z., Zhang, W. & Yin, Y. Understanding the Mott insulating state in 1T-TaS₂ and 1T-TaSe₂. *AAPPS Bull.* **32**, 20 (2022).
3. Bu, K. *et al.* Possible strain induced Mott gap collapse in 1T-TaS₂. *Commun. Phys.* **2**, 1–7 (2019).
4. Adler, R., Kang, C.-J., Yee, C.-H. & Kotliar, G. Correlated materials design: prospects and challenges. *Rep. Prog. Phys.* **82**, 012504 (2018).
5. Vollhardt, D. Dynamical mean-field theory for correlated electrons. *Ann. Phys.* **524**, 1–19 (2012).
6. Kotliar, G. & Vollhardt, D. Strongly Correlated Materials: Insights From Dynamical Mean-Field Theory. *Phys. Today* **57**, 53–59 (2004).
7. Choi, Y. *et al.* Electronic correlations in twisted bilayer graphene near the magic angle. *Nat. Phys.* **15**, 1174–1180 (2019).
8. Zhang, W. *et al.* Visualizing the evolution from Mott insulator to Anderson insulator in Ti-doped 1T-TaS₂. *Npj Quantum Mater.* **7**, 1–8 (2022).
9. Battisti, I. *et al.* Universality of pseudogap and emergent order in lightly doped Mott insulators. *Nat. Phys.* **13**, 21–25 (2017).
10. Yan, L. *et al.* Synthesis and Local Probe Gating of a Monolayer Metal–Organic Framework. *Adv. Funct. Mater.* **31**, 2100519 (2021).
11. Lobo-Checa, J. *et al.* Ferromagnetism on an atom-thick and extended 2D-metal-organic framework. Preprint at <https://doi.org/10.48550/arXiv.2209.14994> (2023).
12. Kumar, D. *et al.* Manifestation of Strongly Correlated Electrons in a 2D Kagome Metal–Organic Framework. *Adv. Funct. Mater.* **31**, 2106474 (2021).
13. Urgel, J. I. *et al.* Controlling Coordination Reactions and Assembly on a Cu(111) Supported Boron Nitride Monolayer. *J Am Chem Soc* **137**, 2420–2423 (2015).
14. Auwärter, W. Hexagonal boron nitride monolayers on metal supports: Versatile templates for atoms, molecules and nanostructures. *Surf. Sci. Rep.* **74**, 1–95 (2019).
15. Field, B., Schiffrin, A. & Medhekar, N. V. Correlation-induced magnetism in substrate-supported 2D metal-organic frameworks. *Npj Comput. Mater.* **8**, 1–10 (2022).
16. Kumar, A., Banerjee, K., Foster, A. S. & Liljeroth, P. Two-Dimensional Band Structure in Honeycomb Metal–Organic Frameworks. *Nano Lett.* **18**, 5596–5602 (2018).
17. Hernández-López, L. *et al.* Searching for kagome multi-bands and edge states in a predicted organic topological insulator. *Nanoscale* **13**, 5216–5223 (2021).
18. Kumar, D., Hellerstedt, J., Lowe, B. & Schiffrin, A. Mesoscopic 2D molecular self-assembly on an insulator. *Nanotechnology* **34**, 205601 (2023).
19. Chen, Y. *et al.* Strong correlations and orbital texture in single-layer 1T-TaSe₂. *Nat. Phys.* **16**, 218–224 (2020).
20. Zhang, Q. *et al.* Tuning Band Gap and Work Function Modulations in Monolayer hBN/Cu(111) Heterostructures with Moiré Patterns. *ACS Nano* **12**, 9355–9362 (2018).

REVIEWER COMMENTS

Reviewer #1 (Remarks to the Author):

The authors have convincingly answered all my questions. The work is more clearly presented and well supported with new additional experimental results and theoretical simulations. I therefore recommend it for publication in Nature Communications.

Minor typos:

Line 175: Fig. S14 Fig. S19 or Section S14

Line 201: Fig. 3c, 3b

Reviewer #2 (Remarks to the Author):

I would like to thank the authors for their efforts in answering my questions, but I am afraid that the answers have not been convincing enough.

In the paper the authors say that they have produced a Mott insulator in a MOF layer. Experimentally, they observe a gap at the Fermi level, depending on the area of the sample (moiré) in which they make the measurements the gap moves in energy following the expected values for the surface potential (not measured) and in some areas the presence of the gap depends on the tip sample position. The assignment of the gap a Mott gap is based only on the DMFT calculations. The authors, in their response to the referees, acknowledge that the DMFT calculations are not capable of reproducing some of the experimental measurements due to limitations in the calculations. Some of the features not explained by the calculations are very prominent in the experiments. Due to these limitations, it seems to me that it is very risky to base the main finding of the article on theoretical calculation that only reproduces the experimental data partially and in some areas of the sample.

According to the authors “the DMFT calculations assume a uniform chemical potential for an infinite system, omitting effects of locality; this assumption is reasonable for the Mott insulating phase (i.e. localized states)”. I think this assumption is wrong, the Mott insulator occurs because itinerant

electrons are located by electronic correlations, so the calculation cannot be local, although the result is that the conduction electrons end up localized and prevented from moving.

Finally, I don't understand why the authors use the existence of DBTJ in some cases and not in others. The MOF is deposited in an insulator on Cu(111), and the MOFF presents a gap at the Fermi level, according to the authors a Mott gap, therefore I do not understand why in some cases it is considered that a DBJT exists and in other cases This is completely ignored without explanation.

Due to all the above, I cannot support the publication of the manuscript.

Reviewer #3 (Remarks to the Author):

The authors have incorporated the suggested comments and made necessary modifications to the manuscript. Now, I recommend its publication.

We thank all three reviewers for their comments, which we address below:

Reviewer #1

The authors have convincingly answered all my questions. The work is more clearly presented and well supported with new additional experimental results and theoretical simulations. I therefore recommend it for publication in Nature Communications.

Minor typos:

Line 175: Fig. S14 Fig. S19 or Section S14

Line 201: Fig. 3c 3b

Author reply:

We thank Reviewer #1 for their positive feedback. We have now corrected these typos.

Reviewer #2

I would like to thank the authors for their efforts in answering my questions, but I am afraid that the answers have not been convincing enough.

In the paper the authors say that they have produced a Mott insulator in a MOF layer. Experimentally, they observe a gap at the Fermi level, depending on the area of the sample (moiré) in which they make the measurements the gap moves in energy following the expected values for the surface potential (not measured) and in some areas the presence of the gap depends on the tip sample position. The assignment of the gap a Mott gap is based only on the DMFT calculations. The authors, in their response to the referees, acknowledge that the DMFT calculations are not capable of reproducing some of the experimental measurements due to limitations in the calculations. Some of the features not explained by the calculations are very prominent in the experiments. Due to these limitations, it seems to me that it is very risky to base the main finding of the article on theoretical calculation that only reproduces the experimental data partially and in some areas of the sample

It is well established that the kagome lattice can host a Mott insulating phase at half-filling if there is significant on-site Coulomb repulsion U (see Ref. 42 in main text). Furthermore, previous theoretical literature supports the existence of a large U and the emergence of a Mott insulating phase in the specific DCA_3Cu_2 MOF that we study here (see Refs. 27, 36 in main text). Additionally, evidence of localised electrons as a result of a large U has already been demonstrated experimentally for this specific 2D MOF (Ref. 25). In this context, the claim of a Mott insulating phase is not controversial – especially given the strong agreement between experiment and dynamical mean-field theory (DMFT) calculations. DMFT is a well-established method for capturing effects of electronic correlations, faithfully describing the Mott metal-insulator transition¹⁻³ (see Refs. 38-41 in main text).

The agreement between experimental dI/dV spectra for the MOF at the moiré pore region and the DMFT spectral functions for the MOF in the Mott insulating phase is excellent, with quantitatively consistent spectral features, including: (i) spectral line shape with lower (LHB) and upper (UHB) Hubbard bands and a ~ 200 meV gap at the Fermi level, and (ii) energy

modulation of the LHB and UHB due to electrostatic potential variations (in the experiment due to the local work function variation given by the hBN/Cu(111) moiré pattern, and in DMFT due to the variation of the chemical potential; Fig. 3b, d, f, g in main text). This energy modulation of the LHB and UHB is perfectly consistent with the moiré local work function variation, regardless of the hBN/Cu(111) moiré pattern periodicity (Supplementary Section S15). This moiré local work function variation of hBN/Cu(111), and its effect on the energy level alignment of atomic and molecular adsorbates, is very well established (Refs. 29-31, 44-47).

This quantitative agreement provides compelling evidence that the DMFT calculations capture the fundamental electronic properties of the 2D DCA₃Cu₂ MOF at the hBN/Cu(111) moiré pore regions, and that the energy gap $E_g = \sim 200$ meV observed experimentally at the Fermi level for the MOF at these moiré pore regions can be attributed to a Mott insulating gap. As stated above, a Mott insulating phase for the kagome lattice and for the specific DCA₃Cu₂ MOF studied here is well supported by previous literature (Refs. 25, 27, 36, 42 in main text).

The DMFT calculations also provide an explanation for the increase in Fermi-level dI/dV signal and absence of an energy gap at the Fermi level for the MOF at the moiré wire regions (for an intermediate tip-sample distance range), where the local work function is larger – and hence the population of the MOF electronic states can be reduced – in comparison with the moiré pore regions. Indeed, the DMFT calculations (main text Fig. 3d, Supplementary Fig. S2b) show that a reduction in the population of the MOF electronic states (i.e., reduction of the chemical potential) leads to a transition from the Mott insulating phase to a metallic phase, with an increase in Fermi-level spectral function and absence of a gap at the Fermi level. That is, the increase in Fermi-level dI/dV signal and absence of energy gap at the Fermi-level is indicative of a MOF metallic phase at the moiré wire regions (for a specific intermediate tip-sample distance range), with electronic states that can be more delocalised than those of the MOF in the Mott insulating phase at the moiré pore regions.

We acknowledge that the effects of the long-range moiré electrostatic potential and of the finite double-barrier tunnel junction (DBTJ) cross section on these arguably delocalised metallic MOF states at moiré wire regions could result in dI/dV features that are not captured by the DMFT calculations (performed for a perfectly periodic, infinite, flat system). Further dI/dV features not captured by DMFT (e.g., dI/dV peaks at positive bias voltage in Figs. 3b, 4d of main text and Supplementary Fig. S24) can also be explained by the susceptibility of tip-induced charging of MOF electronic states at (and in proximity of) the moiré wire regions due to the DBTJ effect. It is important to note that the DBTJ effect does manifest itself also in the moiré pore regions, similar to the moiré wire regions, also leading to energy shifts of the LHB maximum (LHBM) and UHB minimum (UHBM) as the tip-sample distance changes (Supplementary Sections 20, 21). However, given the smaller local work function in comparison to moiré wire regions (Fig. 3c in main text), the MOF energy gap at these moiré pore regions is centred with respect to the Fermi level (Fig. 3b, d in main text), with a significant energy difference between Fermi level and LHBM. This makes the Mott insulating phase at the moiré pore regions robust to energy level shifts given by tip-sample distance changes (within the range of tip-sample distances considered in our study), without tip-induced charging. These factors provide a plausible explanation of why experimental dI/dV spectra and DMFT spectral functions show some differences for the moiré wire regions.

It is important to note that calculations based on DMFT – or even on other theoretical formalisms that capture electron-electron interactions and many-body physics less accurately

(e.g., DFT+U) – on systems with large unit cells [such as the moiré unit cell of our $DCA_3Cu_2/hBN/Cu(111)$ system studied here] are computationally challenging, if not intractable. Such calculations become even more complex or unfeasible if tip-induced effects are taken into account. Our work provides a tractable approach in which electronic correlations and many-body effects are accounted for reliably, with excellent agreement between experiment and theory for moiré pore regions, and good qualitative agreement for the near-Fermi spectral features at the moiré wire regions. We assert that these aspects provide sufficient evidence for: (i) a Mott insulating phase for the MOF at the moiré pore region, regardless of tip-sample distance, (ii) a Mott insulating phase for the MOF at the moiré wire region for large tip-sample distances (i.e., where the DBTJ effect is negligible), and (iii) a metallic phase at the moiré wire region for a specific reduced tip-sample distance range (i.e., where the DBTJ effect leads to a reduction in the population of MOF electronic states). This is the main message of our manuscript. Our study is not trying to say anything further on the nature of the quantum ground state other than classifying it as metallic or insulating. A quantitative reproduction based on DMFT of our experimental dI/dV spectra for all experimental variables (e.g., wide energy range including energies far away from the Fermi level; all locations with respect to $hBN/Cu(111)$ moiré pattern; wide range of tip-sample distances) is beyond the scope of our work.

We have now updated our manuscript to clarify these points (lines 222-225 on p. 9, and lines 307-317 on p. 12-13 in the main text)

According to the authors “the DMFT calculations assume a uniform chemical potential for an infinite system, omitting effects of locality; this assumption is reasonable for the Mott insulating phase (i.e. localized states)”. I think this assumption is wrong, the Mott insulator occurs because itinerant electrons are located by electronic correlations, so the calculation cannot be local, although the result is that the conduction electrons end up localized and prevented from moving.

We acknowledge that this statement could be clearer.

Dynamical mean-field theory (DMFT) is a well-established – widely regarded as the gold-standard – method for understanding Mott metal-insulator transitions. It is not a local method. It fully describes both the Mott insulating and metallic phases, and is responsible for the fundamental understanding of strongly correlated metals¹⁻³ (see Refs. 38-41 in main text).

Let us give a little more technical detail of how DMFT works. Effects of electronic correlations can be accounted for via the self-energy, which in general is nonlocal (i.e., depends on wavevector k), but also intractable to calculate in almost all cases. DMFT calculations assume a self-energy which is local (i.e., independent of k): this makes the calculation of the self-energy feasible. The resulting interacting Green's function is still nonlocal, with local (on-site) correlation effects being fully accounted for (and nonlocal correlation effects being ignored). DMFT calculations do not omit the itinerant nature of electrons; DMFT calculations can still result in metallic phases with itinerant electrons. It is well established that DMFT with such a local approximation of the self-energy captures electronic correlations explicitly, and describes the Mott metal-insulator transition faithfully (see Refs. 38-41 in main text).

Our DMFT calculations assume an infinite, perfectly crystalline, defect-free 2D DCA_3Cu_2 MOF with a uniform chemical potential E_F . Indeed, as stated above, DMFT calculations on systems with large unit cells – such as the long-range moiré unit cell of DCA_3Cu_2 on

hBN/Cu(111) – are computationally challenging, if not intractable. Approximating the potential at each point in space as being uniform is similar in spirit to (but much more accurate than) the local density approximation (LDA) in DFT, where the charge density is locally assumed to be homogeneous. Nevertheless, the LDA can still describe inhomogeneous systems.

Now, for $E_F = 0.25$ to 0.5 eV in Supplementary Fig. S2b (corresponding to half-filling of the 2D kagome system), our DMFT calculations of the MOF spectral function indicate a Mott insulating phase, with a Mott gap $E_g = \sim 200$ meV (for an on-site Coulomb repulsion energy $U = 0.65$ eV; see also k -resolved spectral function for $E_F = 0.4$ eV in Supplementary Fig. S5). Because this Mott phase is stable over this range of chemical potentials, it is robust to significant variations of the chemical potential, up to 0.25 eV.

Furthermore, in this Mott insulating phase, electronic states are localised at the kagome sites, confined within areas that are small in length compared to the distance between nearest-neighbour kagome sites.⁴

In our experiments, the periodicity of the hBN/Cu(111) moiré domains considered ($\lambda > 5$ nm) is significantly larger than the distance between nearest-neighbour kagome sites (~ 1 nm). This moiré pattern imposes to the MOF a periodic modulation of the local work function, with a peak-to-peak modulation amplitude of ~ 0.2 eV for a modulation periodicity $\lambda \approx 12.5$ nm (see Fig. 3 in main text). The amplitude of this local work function modulation becomes smaller with decreasing λ (Supplementary Fig. S22). That is, the MOF is exposed to a periodic modulation of the local electrostatic potential, which varies ‘slowly’ across the molecular kagome lattice.

So, if the MOF is in the Mott insulating phase, the effect of such long-range electrostatic modulation on the localised electronic states is to shift the energy of these localised states accordingly: as long as the electrostatic modulation amplitude does not reach a critical value for the transition to the metallic phase, there is no other dramatic qualitative effect on such localised electronic states. This is consistent with the DMFT-calculated spectral functions in Supplementary Fig. S2b, where the LHB and UHB shift in energy when E_F is varied between 0.25 and 0.5 eV, without other significant qualitative changes. This is also consistent with the experimental dI/dV spectra for the moiré pore regions in Fig. 3b of the main text, where the energy of the LHB and UHB is modulated following the moiré variation in local work function.

In other words, the Mott insulating phase is insensitive to the modulation of the local electrostatic potential, as long as the amplitude of this modulation remains below the threshold for the transition to the metal phase, and as long as the periodicity of this modulation is larger than the distance between nearest-neighbour kagome sites. Importantly, this means that the DMFT calculations for an infinite system in the Mott insulating phase capture the experimental phenomena observed locally at the moiré pore (regardless of tip-sample distance) and wire (for large tip-sample distances) regions.

We have now clarified this by updating lines 639-671 (p. 26, 27) in the Methods section of the main text.

Finally, I don't understand why the authors use the existence of DBTJ in some cases and not in others. The MOF is deposited in an insulator on Cu(111), and the MOF presents a gap at the Fermi level, according to the authors a Mott gap, therefore I do not understand why in some

cases it is considered that a DBJT exists and in other cases This is completely ignored without explanation.

We have revised our manuscript to more clearly explain the effect of the DBTJ on all our experimental measurements (lines 290-295 on p. 12, and lines 697-703 on p. 29 in main text).

Briefly, the DBTJ effect is intrinsic to and manifests itself in all STM measurements of the DCA_3Cu_2 MOF on hBN/Cu(111) system, with energy level shifts due to tip-sample distance variations regardless of the location on the sample (i.e., both at moiré wire and pore regions).

We argue that the DBTJ affects the MOF dI/dV spectra more strongly at the moiré wire regions than at the pore regions due to the larger local work function (Fig. 3c), which leads to a MOF LHBM which is closer to the Fermi level, with the MOF electronic states prone to depopulating and the LHBM energy level susceptible of charging as the tip-sample-distance is reduced. This depopulation of the MOF electronic states results in the transition from Mott insulating (with an energy gap at the Fermi level) to metal phase (with no gap at the Fermi level and an increase in Fermi-level dI/dV signal). This is stated explicitly on lines 274-297 (p. 11-12) of the main text.

As shown in Supplementary Sections S20 and S21, the DBTJ effect also manifests itself at the moiré pore regions, with energy level shifts of the LHBM and UHBM as a function of tip-sample distance, similar to the moiré wire regions, and consistent with the DBTJ model. Due to the smaller local work function at these moiré pore regions (Fig. 3c), the Fermi level lies close to the centre of the MOF energy gap, further from the LHBM in comparison to the moiré wire regions (Fig. 3b, d). That is, the LHBM and UHBM at the moiré pore regions shift in energy as the tip-sample distance is reduced, yet these energy shifts do not lead to a depopulation of MOF electronic states or to a Mott insulator-to-metal transition (for the range of tip-sample distances considered). This is fully consistent with the DMFT calculations of MOF spectral functions in Supplementary Figs. S2b and S4, which show a Mott insulating phase for chemical potentials E_F between ~ 0.25 to ~ 0.5 eV. This means that the Mott insulating phase can be robust to energy level shifts significantly larger than the energy level shifts due to the DBTJ effect (for the range of tip-sample distances considered; see Supplementary Section S4).

In other words, the DBTJ effect manifests itself at both moiré wire and pore regions, but given the local work function difference between these two types of regions, only at the wire regions the electronic properties of the MOF are dramatically altered (i.e., transition from Mott insulator to metallic phase) as the result of such an effect (within the range of tip-sample distances considered in our study).

Due to all the above, I cannot support the publication of the manuscript.

We hope that our comments above have addressed Reviewer #2's concerns.

Reviewer #3

The authors have incorporated the suggested comments and made necessary modifications to the manuscript. Now, I recommend its publication.

We thank Reviewer #3 for their positive feedback.

References

- 1 *Georges, A., Kotliar, G., Krauth, W. & Rozenberg, M. J. Dynamical mean-field theory of strongly correlated fermion systems and the limit of infinite dimensions. Rev Mod Phys 68, 13-125 (1996).*
- 2 *Kotliar, G. & Vollhardt, D. Strongly correlated materials: Insights from dynamical mean-field theory. Phys Today 57, 53-59 (2004).*
- 3 *Kotliar, G. et al. Electronic structure calculations with dynamical mean-field theory. Rev Mod Phys 78, 865-951 (2006).*
- 4 *Fazekas, P. Lecture notes on electron correlation and magnetism. Vol. Volume 5 (WORLD SCIENTIFIC, 1999) isbn:978-981-02-2474-5.*

REVIEWERS' COMMENTS

Reviewer #2 (Remarks to the Author):

I thank the authors for the detail response to my last comments and the corresponding modifications of the manuscript. I consider that they have made an effort to respond in the best possible way to my comments and questions. I consider that the manuscript can be published in its current form.